# De novo learning versus adaptation of continuous control in a manual tracking task

Christopher S Yang[1]*, Noah J Cowan[2], Adrian M Haith[3]

[1]Department of Neuroscience, Johns Hopkins University, Baltimore, United States; [2]Department of Mechanical Engineering, Laboratory for Computational Sensing and Robotics, Johns Hopkins University, Baltimore, United States; [3]Department of Neurology, Johns Hopkins University, Baltimore, United States

**Abstract** How do people learn to perform tasks that require continuous adjustments of motor output, like riding a bicycle? People rely heavily on cognitive strategies when learning discrete movement tasks, but such time-consuming strategies are infeasible in continuous control tasks that demand rapid responses to ongoing sensory feedback. To understand how people can learn to perform such tasks without the benefit of cognitive strategies, we imposed a rotation/mirror reversal of visual feedback while participants performed a continuous tracking task. We analyzed behavior using a system identification approach, which revealed two qualitatively different components of learning: adaptation of a baseline controller and formation of a new, task-specific continuous controller. These components exhibited different signatures in the frequency domain and were differentially engaged under the rotation/mirror reversal. Our results demonstrate that people can rapidly build a new continuous controller *de novo* and can simultaneously deploy this process with adaptation of an existing controller.

***For correspondence:**
christopher.yang@jhmi.edu

**Competing interests:** The authors declare that no competing interests exist.

## Introduction

In many real-world motor tasks, skilled performance requires us to continuously control our actions in response to ongoing external events. For example, remaining stable on a bicycle depends on being able to rapidly respond to the tilt of the bicycle as well as obstacles in our path. The demand for continuous control in such tasks can make it challenging to initially learn them. In particular, new skills often require us to learn arbitrary relationships between our actions and their outcomes (like moving our arms to steer or flexing our fingers to brake). Learning such mappings is thought to depend on the use of time-consuming cognitive strategies (*McDougle et al., 2016*), but continuous control tasks require us to produce responses rapidly, leaving little time for deliberation about our actions. Exactly how we are able to learn new, continuous motor skills therefore remains unclear.

Mechanisms of motor learning have often been studied by examining how people learn to compensate for imposed visuomotor perturbations. Prior studies along these lines have revealed a variety of different ways in which humans learn new motor tasks (*Krakauer et al., 2019*). One of the most well characterized is adaptation, an implicit, error-driven learning mechanism by which task performance is improved by using sensory prediction errors to recalibrate motor output (*Figure 1A*; *Mazzoni and Krakauer, 2006*; *Tseng et al., 2007*; *Shadmehr et al., 2010*). Adaptation is primarily characterized by the presence of aftereffects (*Redding and Wallace, 1993*; *Shadmehr and Mussa-Ivaldi, 1994*; *Kluzik et al., 2008*) and is known to support learning in a variety of laboratory settings including reaching under imposed visuomotor rotations (*Krakauer et al., 1999*; *Fernandez-Ruiz et al., 2011*; *Morehead et al., 2015*), displacements (*Martin et al., 1996*; *Fernández-Ruiz and Díaz, 1999*) or force fields (*Lackner and Dizio, 1994*; *Shadmehr and Mussa-*

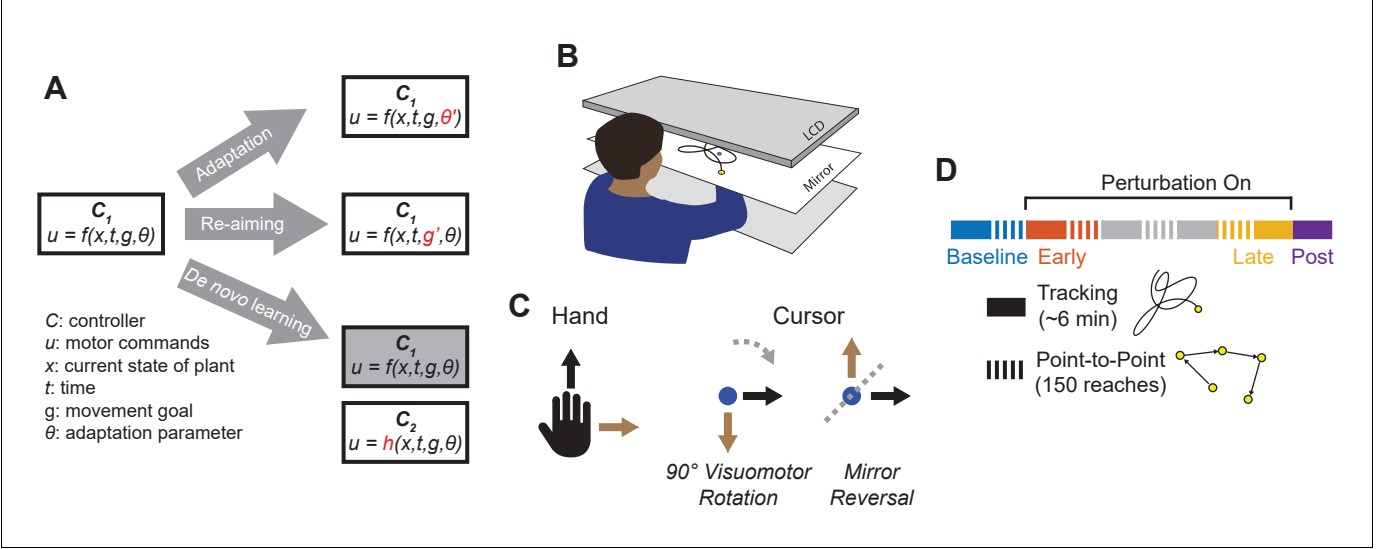

**Figure 1.** Conceptual overview and experimental design. (A) We conceptualize adaptation as a parametric change to an existing controller (changing $\theta$ to $\theta'$), re-aiming as feeding surrogate movement goals to an existing controller (changing $g$ to $g'$), and *de novo* learning as building a new controller ($h$) to replace the baseline controller ($f$). (B) Participants performed planar movements with their right hand while a target (yellow) and cursor (blue) were presented to them on an LCD display. Participants were asked to either move the cursor to a static target (point-to-point task) or track a moving target with the cursor (tracking task). (C) Twenty participants learned to control the cursor under one of two visuomotor perturbations: a 90° clockwise visuomotor rotation ($n = 10$), or a mirror reversal ($n = 10$). (D) Participants alternated between point-to-point reaching (one block = 150 reaches) and tracking (one block = 8 trials lasting 46 s each) in a single testing session in one day. We first measured baseline performance in both tasks under veridical visual feedback (blue), followed by interleaved tracking and point-to-point blocks with perturbed visual feedback from early learning (orange) to late learning (yellow). Blocks between early and late learning are indicated in grey. At the end of the experiment, we assessed aftereffects in the tracking task by removing the perturbation (purple).

*Ivaldi, 1994*), walking on split-belt treadmills (*Choi and Bastian, 2007*; *Finley et al., 2015*), and can also occur in more complex settings such as path integration in gain-altered virtual reality (*Tcheang et al., 2011*; *Jayakumar et al., 2019*). However, it appears that adaptation can only adjust motor output to a limited extent; in the case of visuomotor rotations, implicit adaptation can only alter reach directions by around 15–25°, even when much larger rotations are applied (*Taylor et al., 2010*; *Fernandez-Ruiz et al., 2011*; *Taylor and Ivry, 2011*; *Bond and Taylor, 2015*). Furthermore, under more drastic perturbations, such as a mirror reversal of visual feedback, adaptation seems to act in the wrong direction and can worsen performance rather than improve it (*Abdelghani et al., 2008*; *Hadjiosif et al., 2021*). Thus, other learning mechanisms besides adaptation seem to be required when learning to compensate for perturbations that impose significant deviations from one's existing baseline motor repertoire.

Another way people learn to compensate for visuomotor perturbations is by using re-aiming strategies. This involves aiming one's movements towards a surrogate target rather than the true target of the movement (*Figure 1A*). In contrast to adaptation, in which the controller itself is altered to meet changing task demands, re-aiming feeds an existing controller a fictitious movement goal in order to successfully counter the perturbation without needing to alter the controller itself. It has been shown that people use re-aiming strategies, often in tandem with adaptation, to compensate for visuomotor rotations (*Mazzoni and Krakauer, 2006*; *de Rugy et al., 2012*; *Taylor et al., 2014*; *Morehead et al., 2015*) as well as for imposed force fields (*Schween et al., 2020*) and perturbations to muscular function (*de Rugy et al., 2012*). In principle, re-aiming enables people to compensate for arbitrary visuomotor re-mappings of their environment, including large (90°) visuomotor rotations (*Bond and Taylor, 2015*) or mirror-reversed visual feedback (*Wilterson and Taylor, 2019*). However, implementing re-aiming is a cognitively demanding and time-consuming process that significantly increases reaction times (*Fernandez-Ruiz et al., 2011*; *Haith et al., 2015*; *Leow et al., 2017*; *McDougle and Taylor, 2019*).

A third possible approach to learning, aside from adaptation or re-aiming, is to build a *new* controller to implement the newly required mapping from sensory input to motor output – a process

that has been termed *de novo* learning (*Figure 1A*; *Telgen et al., 2014*; *Sternad, 2018*). This approach contrasts with adaptation, in which an *existing* controller is parametrically altered, and with re-aiming, in which fictitious movement goals are fed to an *existing* controller to generate a successful movement. Previous studies suggest that learning to counter a mirror reversal of visual feedback may engage *de novo* learning. Learning under a mirror reversal shows a number of important differences from learning under a visuomotor rotation: it does not result in aftereffects when the perturbation is removed (*Gutierrez-Garralda et al., 2013*; *Lillicrap et al., 2013*), it shows offline gains (*Telgen et al., 2014*), and it seems to have a distinct neural basis (*Schugens et al., 1998*; *Maschke et al., 2004*; *Morton and Bastian, 2006*; *Gutierrez-Garralda et al., 2013*). However, these properties would also be expected if participants learned to counter the mirror reversal by simply re-aiming their movements to a different target, as has been suggested by *Wilterson and Taylor, 2019*. It is therefore unclear whether or not people ever compensate for visuomotor perturbations by building a *de novo* controller.

How might one dissociate re-aiming from building a new controller? A key property of re-aiming is that it is cognitively demanding and time-consuming to implement (*Fernandez-Ruiz et al., 2011*; *Haith et al., 2015*; *Leow et al., 2017*). This leads to increased reaction times (*Fernandez-Ruiz et al., 2011*), and performance worsens if reaction times are forced to be shorter (*Fernandez-Ruiz et al., 2011*; *Haith et al., 2015*; *Huberdeau et al., 2019*; *McDougle and Taylor, 2019*). While this may not significantly hamper performance in discrete movement tasks like point-to-point reaching or throwing to a stationary target, in continuous control tasks where one's movement goal is constantly and unpredictably changing, movements to the goal cannot be completely planned in advance. Thus, continuous control tasks may severely limit one's ability to use re-aiming strategies and may not be solvable by the same means as point-to-point tasks. Although several studies have examined learning in continuous control tasks (*Schugens et al., 1998*; *Bock and Schneider, 2001*; *Bock et al., 2001*), these studies used relatively slow-moving targets (<0.35 Hz movement), which could potentially be tracked using intermittent 'catch-up' movements that are strategically planned similar to explicit re-aiming of point-to-point movements (*Craik, 1947*; *Miall et al., 1993a*; *Russell and Sternad, 2001*; *Susilaradeya et al., 2019*). To more strictly limit peoples' ability to rely on re-aiming, it is necessary to consider tasks in which movement goals change more quickly than the time it takes for slow cognitive strategies to be applied.

In the present study, participants learned to counter a mirror reversal of visual feedback in both a point-to-point movement task and a continuous tracking task in which a target moved in a pseudo-random sum-of-sinusoids trajectory (*Figure 1B,C*; *Miall et al., 1993b*; *Kiemel et al., 2006*; *Roth et al., 2011*; *Madhav et al., 2013*; *Sponberg et al., 2015*; *Yamagami et al., 2019*). In the tracking task, the target moved at frequencies up to 2 Hz, much faster than in previous tracking experiments, resulting in a target trajectory that was quick, unpredictable, and unlikely to be trackable while using a re-aiming strategy. In order to achieve good tracking performance, participants instead had to continuously generate movements to track the target. Critically, the sum-of-sines structure of the target motion allowed us to employ a frequency-based system identification approach to characterize changes in participants' motor controllers during mirror-reversal learning. We compared learning in this group to that of a second group of participants that learned to counter a visuomotor rotation, where presumably – unlike mirror reversal – adaptation would contribute to learning.

We hypothesized that if participants learned to counter the mirror reversal via *de novo* learning, then they would be able to successfully track the target despite its rapid and unpredictable nature. If, however, the mirror reversal can only be learned through a re-aiming strategy, then we predicted that participants would have difficulty tracking the target and may have to generate intermittent catch-up movements to pursue the target. We further hypothesized that, under the rotation, participants would parametrically alter their baseline controller via adaptation and would therefore be able to smoothly track the target.

## Results

### Participants learned to compensate for the rotation and mirror reversal but using different learning mechanisms

Twenty participants used their right hand to manipulate an on-screen cursor under either a 90° clockwise visuomotor rotation ($n = 10$) or a mirror reversal ($n = 10$) about an oblique 45° axis (*Figure 1C*). These perturbations were chosen such that, in both cases, motion of the hand in the *x*-axis was mapped to motion of the cursor in the *y*-axis and vice versa. Each group practiced using their respective perturbations by performing a *point-to-point task*, reaching towards stationary targets that appeared at random locations on the screen in blocks of 150 trials (*Figure 1D*). Each participant completed the experiment in a single session in 1 day. We assessed both groups' performance in this task by measuring the error between the initial direction of cursor movement and the direction of the target. For the rotation group, this error decreased as a function of training time and plateaued near 0°, demonstrating that participants successfully learned to compensate for the rotation (*Figure 2A*, upper panel). For the mirror-reversal group, the directional error did not show any clear learning curve (*Figure 2A*, lower panel), but performance was better than would be expected if

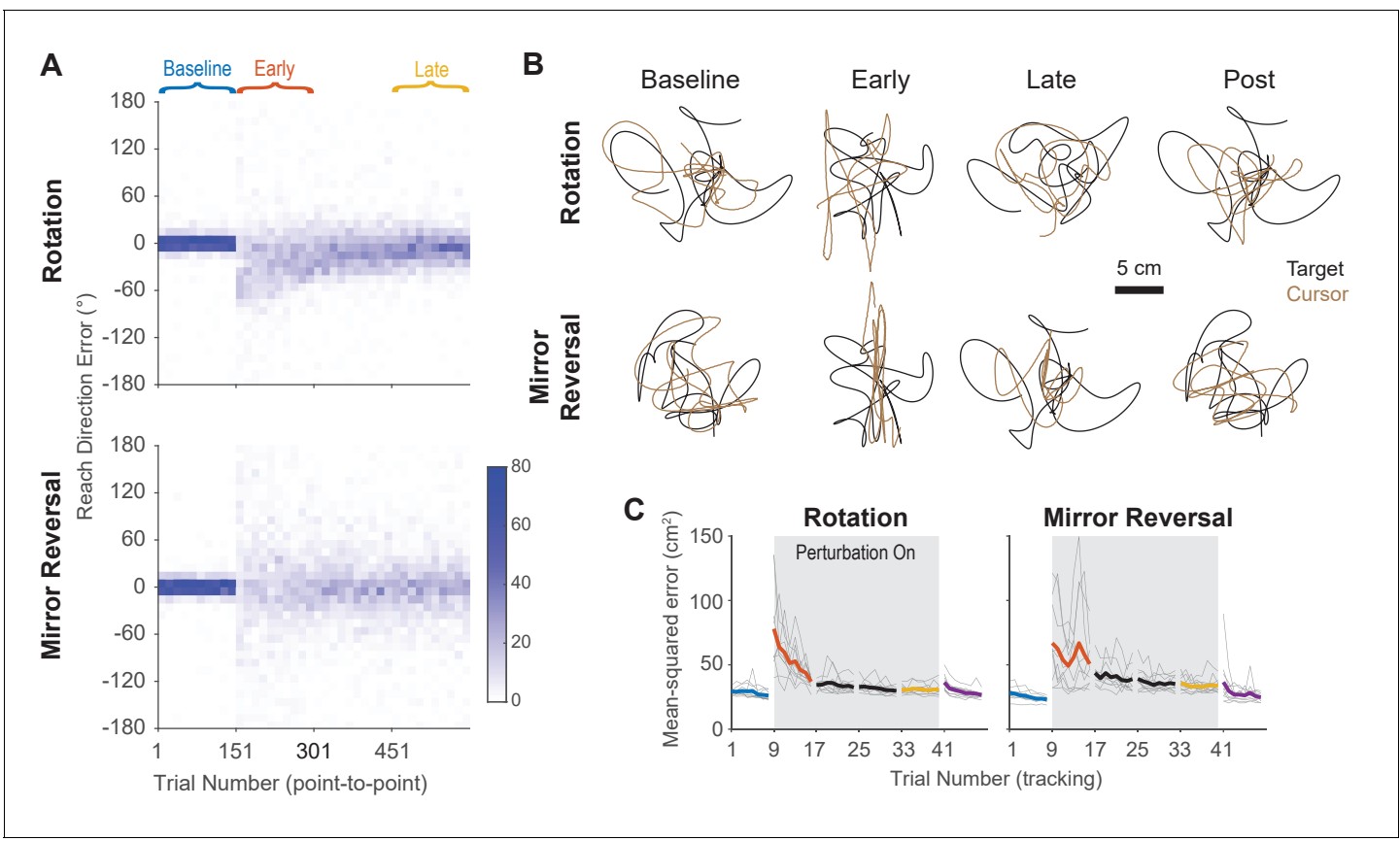

**Figure 2.** Task performance improved in the point-to-point and tracking tasks. (**A**) Performance in the point-to-point task, as quantified by initial reach direction error, is plotted as heat maps for the rotation group (top) and mirror-reversal groups (bottom). Each column shows the distribution of initial reach direction errors, pooled across all participants, over a horizontal bin of 15 trials. The intensity of color represents the number of trials in each 10° vertical bin where the maximum possible value of each bin is 150 (15 trials for 10 participants for each group). (**B**) Example tracking trajectories from a representative participant in each group. Target trajectories are shown in black while cursor trajectories are shown in brown. Each trajectory displays approximately 5 s of movement. (**C**) Performance in the tracking task as quantified by average mean-squared positional error between the cursor and target during each 40 s trial. Individual participants are shown in thin lines and group mean is shown in thick lines.

The online version of this article includes the following video for figure 2:

**Figure 2—video 1.** Video of tracking behavior at different time points during learning.

https://elifesciences.org/articles/62578#fig2video1

participants had not attempted to compensate at all (which would manifest as reach errors uniformly distributed between ±180°). Thus, both groups of participants at least partially compensated for their perturbations in the point-to-point task, consistent with previous findings.

To test whether participants could compensate for these perturbations in a continuous control task after having practiced them in the point-to-point task, we had them perform a manual tracking task. In each 46 s tracking trial (one block = eight trials), participants tracked a target that moved in a continuous sum-of-sinusoids trajectory at frequencies ranging between 0.1 and 2.15 Hz, with distinct frequencies used for *x*- and *y*-axis target movement. The resulting target motion was unpredictable and appeared random. Furthermore, the target's trajectory was altered every block by randomizing the phases of the component sinusoids, preventing participants from being able to learn a specific target trajectory. Example trajectories from single participants are presented in *Figure 2B* (see also *Figure 2—video 1* for a video of tracking behavior).

As an initial assessment of how well participants learned to track the target, we measured the average mean-squared error (tracking error) between the target and cursor positions during every trial (each tracking trial lasted 46 s, 40 s of which was used for analysis; see 'Tracking task' in the Materials and methods for more details). Tracking error improved with practice in both groups of participants, approaching similar levels of error by late learning (*Figure 2C*). Therefore, in both the point-to-point and tracking tasks, participants' performance improved with practice. However, much of this improvement can be attributed to the fact that participants learned to keep their cursor

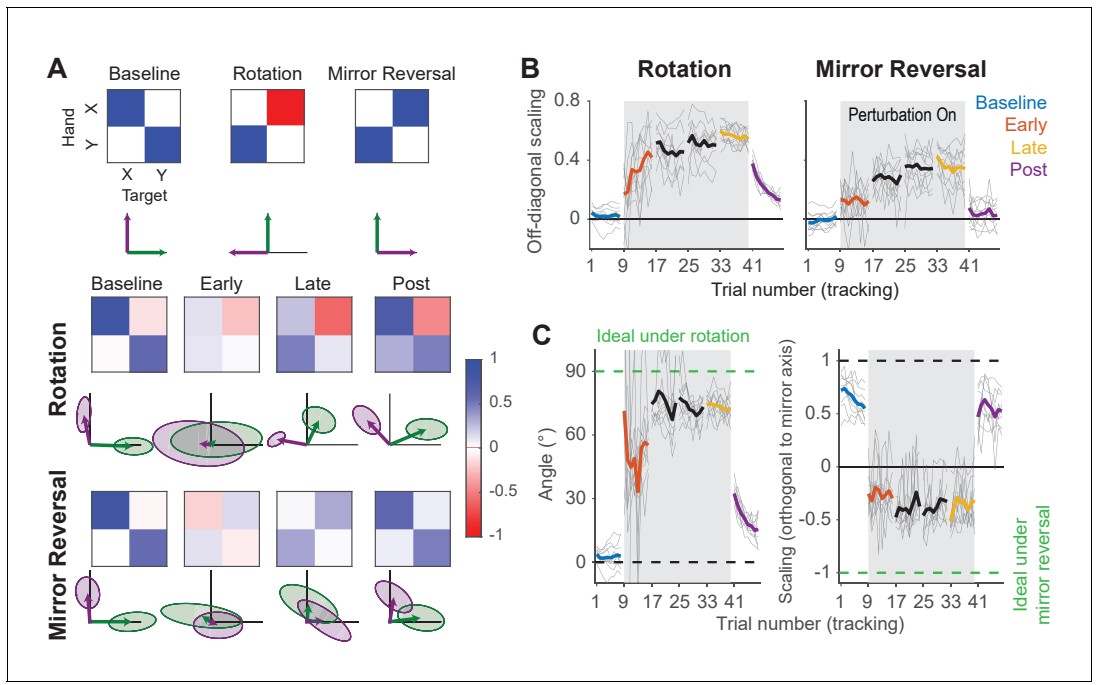

**Figure 3.** The rotation group exhibited reach-direction aftereffects while the mirror-reversal group did not. (**A**) Alignment matrices relating target and hand movement established by trajectory alignment. The top row illustrates the ideal alignment matrices at baseline or to successfully compensate for each perturbation (blue represents positive values, red represents negative values). Alignment matrices (calculated from one trial averaged across participants) from the rotation (middle row) and mirror-reversal (bottom row) groups are depicted at different points during learning. Below each matrix, we visualized how the unit *x* and *y* vectors (black lines) would be transformed by the columns of the matrices (transformed *x* = green, transformed *y* = purple). Shaded areas are 95% confidence ellipses across participants. (**B**) The average of the two off-diagonal elements of the estimated alignment matrices across all blocks of the experiment in the tracking task (for the rotation group, the negative of the element in row 1, column 2 was used for averaging). Grey boxes indicate when the rotation or mirror reversal were applied. Thin black lines indicate individual participants and thick lines indicate the mean across participants. (**C**) (Left: rotation group) Angular compensation for the rotation, computed by approximating each alignment matrix with a pure rotation matrix. (Right: mirror-reversal group) Scaling factor orthogonal to the mirror axis. In each plot, dashed lines depict ideal performance when the perturbation is (green) or is not (black) applied. Thin black lines indicate individual participants and thick lines indicate the mean across participants.

The online version of this article includes the following source data for figure 3:

**Source data 1.** This file contains the results of all statistical analyses performed on the data in *Figure 3B*.

within the bounds of target movement; during early learning, participants' cursors often deviated far outside the area of target movement, thus inflating the tracking error.

To better quantify improvements in participants' ability to track the target, we examined the geometric relationship between hand and target trajectories—an approach that would be more sensitive to the small changes in movement direction associated with rotation/mirror reversal learning, not just large deviations outside the target's movement area. We aligned the hand and target tracking trajectories with a linear transformation matrix (alignment matrix) that, when applied to the target trajectory, minimized the discrepancy between the hand and target trajectories (see Materials and methods for details). This matrix compactly summarizes the relationship between target movements and hand movements and can be thought of as a more general version of reach direction for point-to-point movements. We visualized these matrices by plotting their column vectors (green and purple arrows in *Figure 3A*) which depicts how they would transform the unit $x$ and $y$ vectors.

In *Figure 3A*, we illustrate how ideal performance under different visual feedback conditions would manifest in the alignment matrices and vectors. These matrices should approximate the identity matrix when performing under veridical feedback and similarly approximate the inverse of the applied perturbation matrix under perturbed feedback. Incomplete compensation would manifest as, for example, a 45° counter-clockwise rotation matrix in response to the 90° clockwise rotation. For both groups of participants, the estimated alignment matrices were close to the identity matrix at baseline and approached the inverses of the respective perturbations by late learning (*Figure 3A*), demonstrating that participants performed the task successfully at baseline and mostly learned to compensate for the imposed perturbations.

To test whether these changes were statistically significant, we focused on the off-diagonal elements of the matrices. These elements critically distinguish the different transformations from one another and from baseline. In the last trial of the late-learning block, both the rotation (linear mixed-effects model [see 'Statistics' in Materials and methods for details about the model structure]: interaction between group and block, $F(2, 36) = 7.56$, $p = 0.0018$; Tukey's range test: $p<0.0001$; see *Figure 3—source data 1* for p-values of all statistical comparisons related to *Figure 3*) and mirror-reversal groups (Tukey's range test: $p<0.0001$) exhibited off-diagonal values that were significantly different from the first trial of the baseline block (*Figure 3B*), and in the appropriate direction to compensate for their respective perturbations.

From these matrices, we derived additional metrics associated with each perturbation to further characterize learning. For the rotation group, we computed a compensation angle, $\theta$, using a singular value decomposition approach (*Figure 3C*; see 'Trajectory-alignment analysis' in Materials and methods for details). At baseline, we found that $\theta = 3.8 \pm 1.0°$ (mean ± SEM), and this increased to $\theta = 72.5 \pm 1.9°$ by late learning. For the mirror-reversal group, to assess whether participants learned to flip the direction of their movements across the mirroring axis, we computed the scaling of the target trajectory along the direction orthogonal to the mirror axis (*Figure 3C*). This value was positive at baseline and negative by late learning, indicating that participants successfully inverted their hand trajectories relative to that of the target.

Lastly, we sought to confirm that the rotation and mirror reversal were learned using different mechanisms, as has been suggested by previous studies (*Gutierrez-Garralda et al., 2013*; *Telgen et al., 2014*). We did so by assessing whether participants in each group expressed reach-direction aftereffects – the canonical hallmark of adaptation – at the end of the experiment, following removal of each perturbation in the tracking task (and with participants made explicitly aware of this). Again estimating alignment matrices (*Figure 3B*), we found that the magnitude of aftereffects (as measured by the off-diagonal elements of the alignment matrices) was different between the two groups in the first trial post-learning (Tukey's range test: $p<0.0001$). Within groups, the off-diagonal elements for the rotation group were significantly different between the first trial of baseline and the first trial of post-learning (Tukey's range test: $p<0.0001$), indicating clear aftereffects. These aftereffects corresponded to a compensation angle of $\theta = 32.4 \pm 1.4°$, similar to the magnitude of aftereffects reported for visuomotor rotation in point-to-point tasks (*Bond and Taylor, 2015*; *Morehead et al., 2017*). For the mirror-reversal group, by contrast, the off-diagonal elements from the first trial of post-learning were not significantly different from the first trial of baseline (Tukey's range test: $p = 0.2057$; baseline range: –0.11 to 0.11; post-learning range: −0.07 to 0.28), suggesting negligible aftereffects. The lack of aftereffects under mirror reversal implies that participants did not

counter this perturbation via adaptation of an existing controller and instead used an alternative learning mechanism.

In summary, these data suggest that participants were able to compensate for both perturbations in the more challenging tracking task. Consistent with previous studies focusing on point-to-point movements, these data support the idea that the rotation was learned via adaptation, while the mirror reversal was learned via a different mechanism – putatively, *de novo* learning.

## Participants used continuous movements to perform manual tracking

Although participants could learn to successfully perform the tracking task under the mirror reversal, it is not necessarily clear that they achieved this by building a new, continuous controller; the largest amplitudes and velocities of target movement occurred primarily at low frequencies (0.1–0.65 Hz), which could potentially have allowed participants to track the target through a series of discretely planned 'catch-up' movements (*Craik, 1947*; *Miall et al., 1993a*; *Russell and Sternad, 2001*; *Susilaradeya et al., 2019*) that might have involved re-aiming. If participants were employing such a re-aiming strategy, we would expect this to compromise their ability to track the target continuously. To examine the possibility that participants may have tracked the target intermittently rather than continuously, we turned to linear systems analysis to analyze participants behavior at a finer-grained level than was possible through the trajectory-alignment analysis.

According to linear systems theory, a linear system will always translate sinusoidal inputs into sinusoidal outputs at the same frequency, albeit potentially scaled in amplitude and shifted in phase. Additionally, linearity implies that the result of summing two input signals is to simply sum the respective outputs. Therefore, a linear system can be fully described in terms of how it maps sinusoidal inputs to outputs across all relevant frequencies. If participants' behavior can be well approximated by a linear model – as is often the case for planar arm movements (*McRuer and Jex, 1967*; *Yamagami et al., 2019*; *Zimmet et al., 2020*) – then we can fully understand their tracking behavior in terms of their response to different frequencies of target movement. The design of the tracking task enabled us to examine the extent to which participants' behavior was linear; if participants were indeed behaving linearly (which would suggest they were tracking the target continuously), then we should find that their hand also moved according to a sum-of-sines trajectory, selectively moving at the same frequencies as the target.

We assessed whether participants selectively moved at the same frequencies as target movement by first converting their trajectories to a frequency-domain representation via the discrete Fourier transform. This transformation decomposes the full hand trajectory into a sum of sinusoids of different amplitudes, phases, and frequencies. *Figure 4A* shows the amplitude spectra (i.e., amplitude of movement as a function of frequency) of hand movements in the *x*-axis at different points during the experiment, averaged across participants (analogous data for *y*-axis movements can be found in *Figure 4—figure supplement 1* and data from single subjects can be found in *Figure 4—figure supplement 2*). The amplitudes and frequencies of target movement are shown as diamonds (*x*- and *y*-axis sinusoids in green and brown, respectively) and the amplitude of participants' movements at those same frequencies are marked by circles.

At baseline, late learning, and post-learning, participants moved primarily at the frequencies of *x*- or *y*-axis target movement (*Figure 4A*). At frequencies that the target did *not* move in, the amplitude of hand movement was low. This behavior resulted in clearly discernible peaks in the amplitude spectra, which is consistent with the expected response of a linear system. In contrast, participants' behavior at early learning was qualitatively different, exhibiting high amplitude at movement frequencies below 1 Hz, regardless of whether the target also moved at that frequency. This suggests that a much greater proportion of participants' behavior was nonlinear/noisy, as would be expected during early learning when neither group of participants had adequately learned to counter the perturbations.

As a further test of the linearity of participants' behavior, we computed the spectral coherence between target and hand movement, which is simply the correlation between two signals in the frequency domain. As demonstrated by *Roddey et al., 2000*, for an arbitrary system responding to an arbitrary input, the fraction of the system's response that can be explained by a linear model is proportional to the coherence between the input and the system's output (a perfectly linear, noiseless system would exhibit a coherence of 1 across all frequencies). At baseline, for both groups, we found that the coherence between target movement and participants' hand movement was roughly 0.75 in

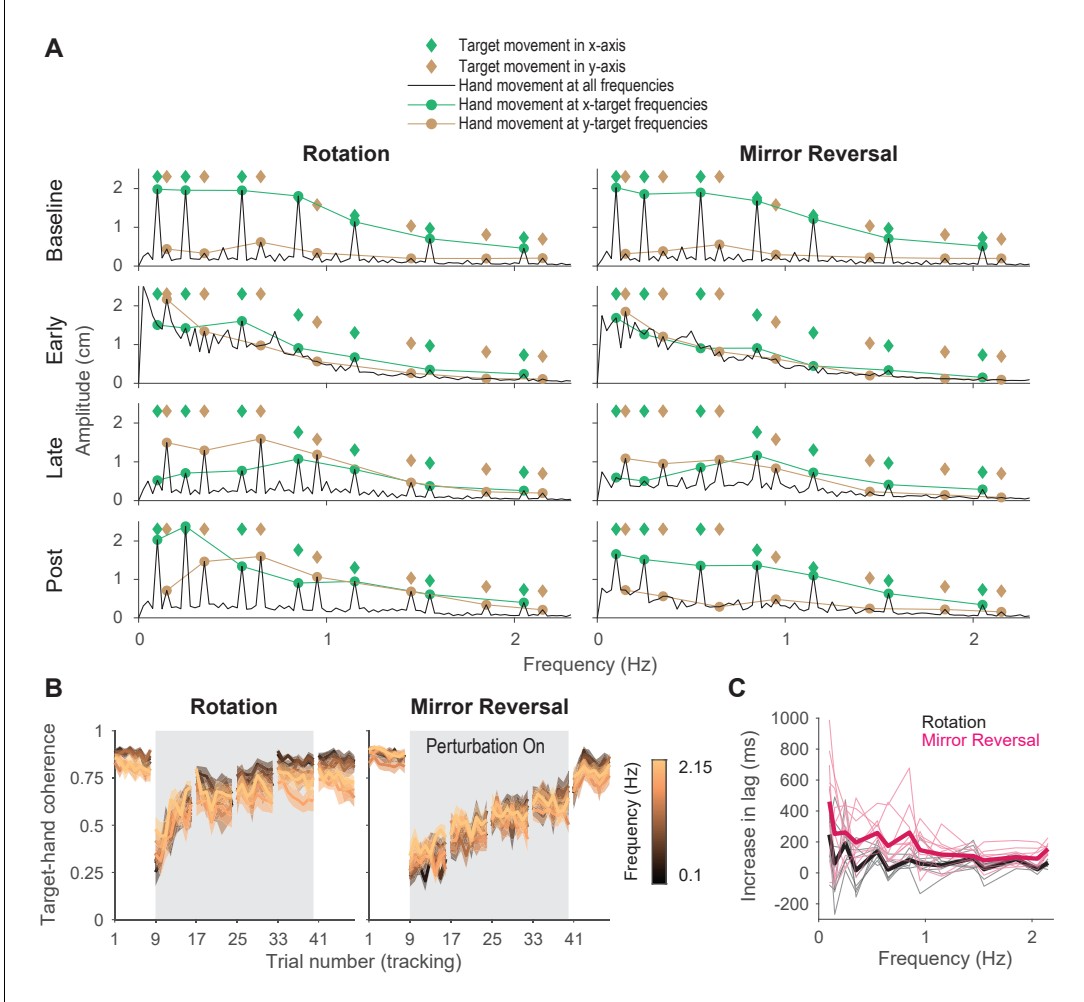

**Figure 4.** Tracking behavior was approximately linear, indicating that the hand tracked the target continuously. (**A**) Amplitude spectra of *x*-axis hand trajectories (black line) averaged across participants from one trial in each listed block. In each plot, the amplitudes and frequencies of target motion are indicated by diamonds (green: *x*-axis target frequencies; brown: *y*-axis target frequencies). Hand responses at *x*- and *y*-axis target frequencies are highlighted as green and brown circles, respectively, and are connected by lines for ease of visualization. (**B**) Spectral coherence between target movement in the *x*-axis and hand movement in both axes. This measure is proportional to the linear component of the hand's response to the target. Darker colors represent lower frequencies and lighter colors represent higher frequencies. Error bars are SEM across participants. (**C**) Difference in phase lag between movements at late learning and baseline. Data from individual participants are shown as thin lines and averages for the rotation (black) and mirror-reversal (pink) groups are shown as thick lines.

The online version of this article includes the following figure supplement(s) for figure 4:

**Figure supplement 1.** Amplitude spectra (*y*-axis), spectral coherence, and phase lag plots.

**Figure supplement 2.** Amplitude spectra of *x*-axis hand movements from single subjects.

both the *x*- and *y*-axes (*Figure 4B*), meaning that 75% of participants' behavior could be accounted for by a linear model. Although dramatically lower during early learning, the coherence approached that of baseline by late learning, indicating that the proportion of participants' behavior that could be accounted for by a linear model increased with more practice time.

As with any correlation, the residual variance in behavior not explained by a linear model was attributable to either nonlinearities or noise. Because catch-up movements could manifest as nonlinear behavior, we estimated the additional variance that could be explained by a nonlinear, but *not* a linear, model by measuring the square root of the coherence between multiple responses to the same input (*Roddey et al., 2000*), that is, hand movements from different trials within a block. We found that across all blocks, only an additional 5–10% of tracking behavior could be explained by a nonlinear model (data not shown), suggesting that most of the residual variance was attributable

to noise and that a linear model was almost as good as a nonlinear model at explaining behavior on a trial-by-trial basis. In summary, these analyses suggest that participants' behavior at baseline, late learning, and post-learning could be well described as a linear system, thereby suggesting that their movements were continuous.

Although behavior was approximately linear across all frequencies, it is possible that performing a sequence of discretely planned catch-up movements – which might have depended on the use of a re-aiming strategy – could approximate linear behavior, particularly at low frequencies of movement. As a result, we analyzed the lag between hand and target movements to examine the plausibility of participants repeatedly re-aiming in the tracking task. Previous work suggests that in tasks with many possible target locations, planning point-to-point movements under large rotations of visual feedback incurs an additional ~300 ms of planning time on top of that required under baseline conditions (*Fernandez-Ruiz et al., 2011*; *McDougle and Taylor, 2019*). In the context of the tracking task, this suggests that, compared to baseline, people would require an additional 300 ms of reaction time for each catch-up movement under the rotation or mirror reversal, which would increase the lag between hand movements relative to the target.

We computed this lag at late learning and baseline at every frequency of target movement (*Figure 4—figure supplement 1C*). We then examined how much this lag increased from baseline to late learning *Figure 4C*. For all but the lowest frequency of movement for the mirror-reversal group, the average increase in lag was below 300 ms. In fact, averaging across all frequencies, the increase in lag for the rotation and mirror-reversal groups were 83 ± 31 and 191 ± 62 ms (mean ± standard deviation across participants), respectively. This analysis suggests that participants responded to target movement quickly—more quickly than would be expected if participants tracked the target by repeatedly re-aiming toward an alternative target location.

In summary, the above analyses show that participants were able to track the target smoothly and continuously after learning to compensate for either the rotation or the mirror reversal. Participants did not appear to be making intermittent catch-up movements nor relying on a re-aiming strategy. Rather, their performance suggests that they were able to continuously track the target by building a *de novo* controller.

## Adaptation and *de novo* learning exhibit distinct signatures in the frequency domain

The fact that tracking behavior could be well approximated as a linear dynamical system, particularly late in learning, facilitates a deeper analysis into how learning altered participants' control capabilities. Following this approach, we treated each 40 s tracking trial as a snapshot of participants' control capabilities at a particular time point during learning, assuming that the behavior could be regarded as being generated by a linear, time-invariant system. Although participants' behavior changed over the course of the experiment due to the engagement of (likely nonlinear) learning processes, within the span of individual trials, our data suggest that their behavior was both approximately linear (*Figure 4A,B*) and changed only minimally from trial-to-trial (*Figure 3B,C*), validating the use of linear systems analysis on single-trial data.

We first examined learning in the amplitude spectra analysis. To perfectly compensate for either the rotation or the mirror reversal, participants' responses to movement of the target in the *x*-axis needed to be remapped from the *x*-axis to the *y*-axis, and vice versa for movement of the target in the *y*-axis. Since the target moved at different frequencies in each axis, this remapping could be easily observed in the amplitude spectra as peaks at different frequencies. During early learning, both groups' movements were nonlinear and were not restricted to *x*- or *y*-axis target frequencies (*Figure 4A*). However, by late learning, both groups learned to produce *x*-axis hand movements in response to *y*-axis target frequencies, indicating some degree of compensation for the perturbation. However, they also inappropriately continued to produce *x*-axis hand movements at *x*-axis target frequencies, suggesting that the compensation was incomplete.

After the perturbation was removed, the rotation group exhibited *x*-axis hand movements at both *x*- and *y*-axis target frequencies, unlike baseline where movements were restricted to *x*-axis target frequencies (*Figure 4A*). The continued movement in response to *y*-axis target frequencies indicated aftereffects of having learned to counter the rotation, consistent with our earlier trajectory-alignment analysis. In contrast, the amplitude spectra of the mirror-reversal group's *x*-axis hand movements post-learning was similar to baseline, suggesting negligible aftereffects and again recapitulating the

findings of our earlier analysis. These features of the amplitude spectra, and the differences across groups, were qualitatively the same for *y*-axis hand movements (*Figure 4—figure supplement 1*) and were also evident in individual subjects (*Figure 4—figure supplement 2*).

Although the amplitude spectra illustrate important features of learning, they do not carry information about the directionality of movements and thus do not distinguish learning of the two different perturbations; perfect compensation would lead to identical amplitude spectra for each perturbation. In order to distinguish these responses, we needed to determine not just the amplitude, but the direction of the response along each axis, i.e., whether it was positive or negative. We used *phase* information to disambiguate the direction of the response (the sign of the gain) by assuming that the phase of the response at each frequency would remain similar to baseline throughout learning. We then used this information to compute signed gain matrices, which describe the linear transformations relating target and hand motion (*Figure 5—figure supplement 1*). These matrices relay similar information as the alignment matrices in *Figure 3* except that here, different transformations were computed for different frequencies of movement. To construct these gain matrices, the hand responses from neighboring pairs of *x*- and *y*-axis target frequencies were grouped together. This grouping was performed because target movement at any given frequency was one-dimensional, but target movement across two neighboring frequencies was two-dimensional; examining hand/target movements in this way thus provided two-dimensional insight into how the rotation/mirroring of hand responses varied across the frequency spectrum (see 'Frequency-domain analysis' in Materials and methods for details).

Similar to the trajectory-alignment analysis, these gain matrices should be close to the identity matrix at baseline the inverse of the matrix describing the perturbation if participants are able to perfectly compensate for the perturbation. We again visualized these frequency-dependent gain matrices by plotting their column vectors, which illustrates the effect of the matrix on the unit *x* and *y* vectors, only now we include a set of vectors for each pair of neighboring frequencies (*Figure 5A*: average across subjects, *Figure 5—figure supplement 2A*: single subjects). We also plotted the same information represented as colormapped gain matrices in *Figure 5—figure supplement 1*, similar to *Figure 3A*.

At baseline, participants in both groups responded to *x*- and *y*-axis target motion by moving their hands in the *x*- and *y*-axes, respectively, with similar performance across all target frequencies. Late in learning for the rotation group, participants successfully compensated for the perturbation – apparent through the fact that all vectors rotated clockwise during learning. The extent of compensation, however, was not uniform across frequencies; compensation at low frequencies (darker arrows) was more complete than at high frequencies (lighter arrows). For the mirror-reversal group, compensation during late learning occurred most successfully at low frequencies, apparent as the darker vectors flipping across the mirror axis (at 45° relative to the *x*-axis) from their baseline direction. At high frequencies, however, responses failed to flip across the mirror axis and remained similar to baseline.

To quantify these observations statistically, we focused again on the off-diagonal elements of the gain matrices from individual trials. The rotation group's gain matrices were altered in the appropriate direction to counter the perturbation, showing a significant difference between the first trial of baseline and the last trial of late learning at all frequencies (*Figure 5B*; linear mixed-effects model (see 'Statistics' in Materials and methods for details about the model structure): interaction between group, block, and frequency, $F(12, 360) = 3.39$, $p = 0.0001$; data split by frequency for post hoc Tukey's range test: Bonferroni-adjusted $p<0.05$ for all frequencies; see *Figure 5—source data 1* for p-values of all statistical comparisons related to *Figure 5*). Comparing the first trial of baseline and last trial of late learning for the mirror-reversal group revealed that the low-frequency gain matrices were also altered in the appropriate direction to counter the perturbation (Tukey's range test: Bonferroni-adjusted $p<0.001$ for lowest three frequencies), but the high-frequency gain matrices were not significantly different from each other (Tukey's range test: $p>0.6$ [not Bonferroni-adjusted] for highest three frequencies; baseline gain range: $-0.18$ to $0.18$; late-learning gain range: $-0.25$ to $0.66$).

Calculating a rotation matrix that best described the rotation group's gain matrix at each frequency (using the same singular value decomposition approach applied to the alignment matrices) revealed that participants' baseline compensation angle was close to 0° at all frequencies (*Figure 5C*). By late learning, compensation was nearly perfect at the lowest frequency but was only

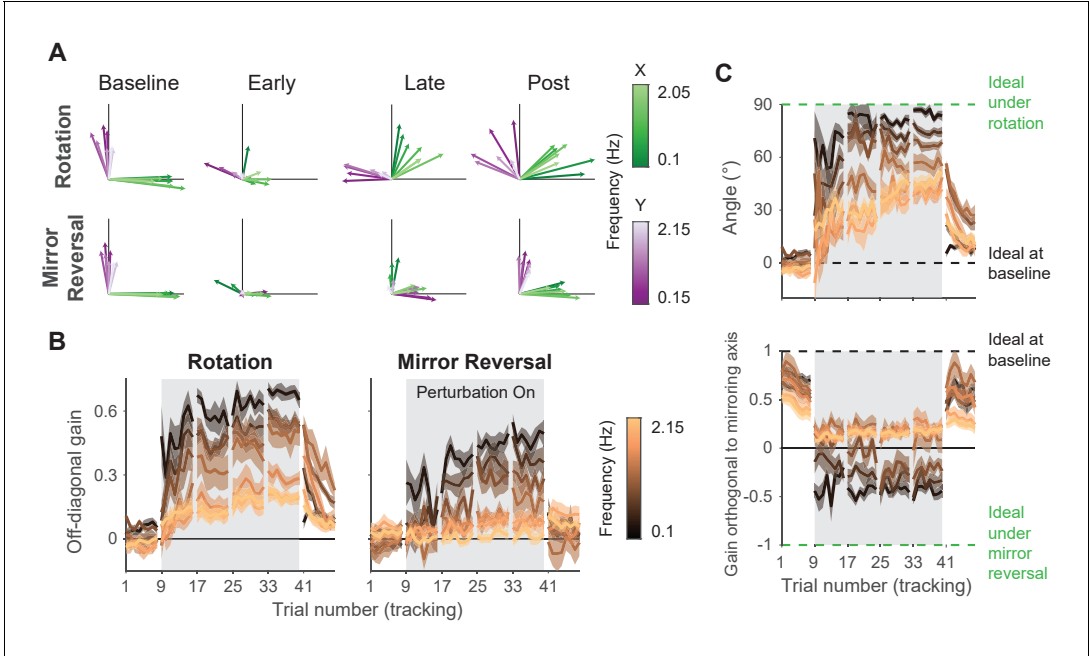

**Figure 5.** Adaptation and *de novo* learning exhibit distinct frequency-dependent signatures. We estimated how participants transformed target motion into hand movement across different frequencies (i.e., gain matrix analysis). (**A**) Visualizations of the gain matrices relating target motion to hand motion across frequencies (associated gain matrices can be found in *Figure 5—figure supplement 1*). These visualizations were generated by plotting the column vectors of the gain matrices from one trial of each listed block, averaged across participants. Green and purple arrows depict hand responses to *x*- and *y*-axis target frequencies, respectively. Darker and lighter colors represent lower and higher frequencies, respectively. (**B**) Average of the two off-diagonal values of the gain matrices at different points during learning. Grey boxes indicate when the rotation or mirror reversal were applied. (**C**) (Top) Compensation angle as a function of frequency for the rotation group. (Bottom) Gain of movement orthogonal to the mirror axis for the mirror-reversal group. Green and black dashed lines show ideal compensation when the perturbation is or is not applied, respectively. All error bars in this figure are SEM across participants.

The online version of this article includes the following source data and figure supplement(s) for figure 5:

**Source data 1.** This file contains the results of all statistical analyses performed on the data in *Figure 5B*.

**Figure supplement 1.** Example gain matrices for each block and frequency.

**Figure supplement 2.** Gain matrix analysis performed on single-subject data.

partial at higher frequencies. For the mirror-reversal group, the gains of participants' low-frequency movements orthogonal to the mirror axis were positive at baseline and became negative during learning, appropriate to counter the perturbation. At high frequencies, by contrast, the gain reduced slightly during learning but never became negative. Thus, both groups of participants were successful at compensating at low frequencies but, at high frequencies, the rotation group was only partially successful and the mirror-reversal group was largely unsuccessful.

The gain matrices also recapitulated the post-learning trends from the trajectory-alignment analysis in *Figure 3*. In the first post-learning trial, the rotation group's off-diagonal gains were significantly different from the first trial of baseline for all frequencies except the lowest (*Figure 5B*; Tukey's range test: Bonferroni-adjusted $p < 0.003$ for highest six frequencies). By contrast, there was no strong evidence that the mirror-reversal group's post-learning matrices were significantly different from baseline (Tukey's range test: $p > 0.04$ (not Bonferroni-adjusted) for all frequencies; baseline gain range: $-0.18$ to $0.18$; post-learning gain range: $-0.49$ to $0.37$). Additionally, the post-learning gains differed significantly between the rotation and mirror-reversal groups, albeit only for three of the intermediate frequencies (Tukey's range test: Bonferroni-adjusted $p < 0.001$). Similar trends were evident in the compensation angles for the rotation group and orthogonal gains for the mirror-reversal group (*Figure 5C*). These data again suggest that the rotation group expressed aftereffects while the mirror-reversal group did not.

To summarize, compensation for the visuomotor rotation was expressed at both low and high frequencies of movement, and this compensation resulted in reach-direction aftereffects of similar

magnitude to that reported in previous studies using point-to-point movements (*Taylor et al., 2010*; *Fernandez-Ruiz et al., 2011*; *Taylor and Ivry, 2011*; *Bond and Taylor, 2015*). This suggests that participants learned to compensate for the rotation through adaptation, that is, by adapting their existing baseline controller. In contrast, the mirror-reversal group only expressed compensation at low frequencies of movement, exhibiting little to no compensation at high frequencies, and did not exhibit aftereffects, suggesting that they did not learn through adaptation of an existing controller. Combined with the results from *Figure 4* suggesting that participants did not utilize a re-aiming strategy while tracking, these data suggest that participants learned to counter the mirror reversal by building a new controller from scratch, that is, through *de novo* learning.

Learning in the rotation group also appeared to be, to some extent, achieved through *de novo* learning. The magnitude of aftereffects in this group (~25°) was only a fraction of the overall compensation achieved (~70°) during late learning, suggesting that implicit adaptation cannot entirely account for the rotation group's behavior. The results from *Figure 4* also suggest that the rotation group's behavior could not be explained by a strategy of tracking the target through a series of re-aimed catch-up movements. Examining the time course of learning for both groups in *Figure 5B*, while the rotation group's gains were overall higher than the mirror-reversal group's, there was a striking similarity in the frequency-dependent pattern of learning between the two groups. We therefore conclude that the residual learning not accounted for by adaptation was attributable to the same *de novo* learning process that drove learning under the mirror reversal.

## Examining the effect of re-aiming strategies on learning

Although the data suggest that participants did not primarily rely on a re-aiming strategy while tracking, participants likely did use such a strategy to learn to counter the rotation/mirror reversal while performing point-to-point reaches. How important might such cognitive strategies be for ultimately learning the tracking task? To better understand this, we performed a follow-up experiment with 20 additional participants. This experiment was similar to the main experiment except for the fact that participants experienced the rotation/mirror reversal almost exclusively in the tracking task, performing only 15 point-to-point reaches between the early and late learning tracking blocks compared to the 450 reaches in the main experiment (*Figure 6A*).

We applied the gain matrix analysis from *Figure 5* to data from this experiment and found that our previous results were largely reproduced despite the very limited point-to-point training (*Figure 6B–D*). The rotation group exhibited aftereffects in the gain matrices (linear mixed-effects model [see 'Statistics' in Materials and methods for details about the model structure]: interaction between block, frequency, and group, $F(12, 360) = 3.26$, $p = 0.0002$; data split by frequency for post hoc Tukey's range test: Bonferroni-adjusted $p<0.01$ for four of seven frequencies; see *Figure 6— source data 1* for p-values of all statistical comparisons related to *Figure 6*) which were significantly greater than that of the mirror-reversal group (Tukey's range test: Bonferroni-adjusted $p<0.0005$ for two out of seven frequencies). In contrast, the mirror-reversal group did not express aftereffects (Tukey's range test: $p>0.4$ [not Bonferroni-adjusted] for all seven frequencies; *Figure 6C*). Furthermore, the rotation group exhibited compensation at high frequencies (Tukey's range test: Bonferroni-adjusted $p = 0.0073$ at third highest frequency) whereas the mirror-reversal group did not (Tukey's range test: $p>0.5$ [not Bonferroni-adjusted] for highest four frequencies). These trends were also evident in single participants (*Figure 6—figure supplement 1*). Thus, the follow-up experiment provided evidence that the effects we observed in the main experiment were replicable.

Directly comparing the results between the two experiments (comparing *Figures 6C* and *5B*), we found that participants in the follow-up experiment exhibited significantly less compensation in the last trial of late learning compared to participants in the main experiment, as quantified by the off-diagonal gain (two-way ANOVA [see 'Statistics' in Materials and methods for details about the ANOVA]: main effect of experiment, $F(1, 252) = 37.69$, $p<0.0001$, with no significant interactions between any predictors; see *Figure 6—source data 1* for more detailed statistics related to *Figure 6*). It is unclear, however, whether this reduced learning was attributable to participants being unable to develop a re-aiming strategy without point-to-point training, or whether it could be explained by the fact that participants simply spent less total time being exposed to the perturbations.

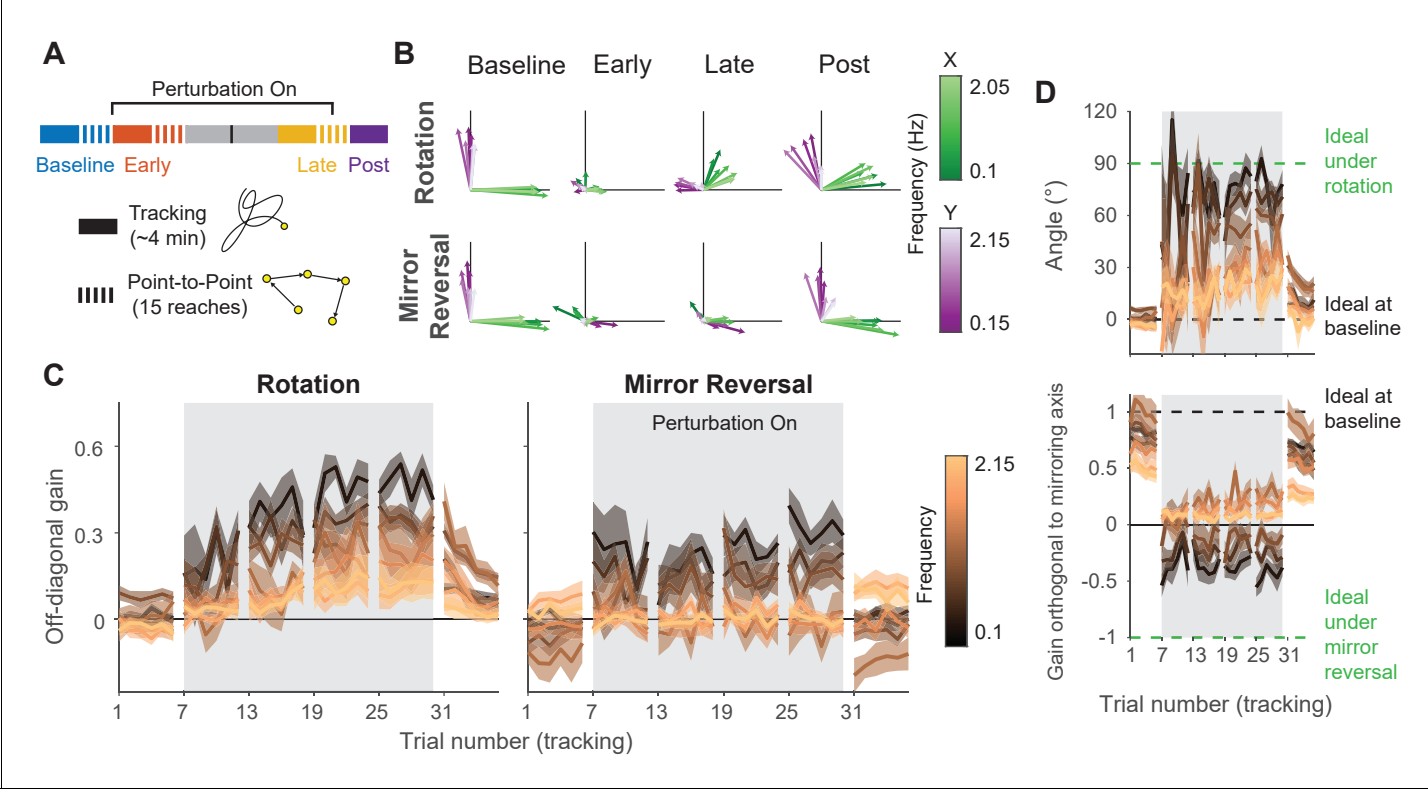

**Figure 6.** Making point-to-point reaches improves tracking performance, especially under mirror reversal. (A) Participants learned to counter either a visuomotor rotation ($n = 10$) or mirror-reversal ($n = 10$). The experimental design was similar to the main experiment except point-to-point reaching practice was almost entirely eliminated; between the early- and late-learning tracking blocks, participants only performed 15 point-to-point reaches. The purpose of these reaches was not for training but simply to assess learning in the point-to-point task. (B–D) Gain matrix analysis, identical to that in *Figure 5*, performed on data from the follow-up experiment. (B) Visualization of the gain matrix from one trial of each listed block, averaged across participants. (C) Off-diagonal elements of the gain matrices, averaged across participants. (D) Computed rotation angle for the rotation group's gain matrices (upper) and gain orthogonal to mirroring axis for the mirror-reversal group (lower), averaged across participants. All error bars in this figure are SEM across participants.

The online version of this article includes the following source data and figure supplement(s) for figure 6:

**Source data 1.** This file contains the results of all statistical analyses performed on the data in *Figure 6C*.
**Figure supplement 1.** Gain matrix analysis performed on single-subject data for the follow-up experiment.

Therefore, while virtually eliminating point-to-point training may have diminished participants' ability to learn the task, participants were still able to counter the perturbation to some extent, reproducing the most salient findings from the main experiment.

## Discussion

In the present study, we tested whether participants could learn to successfully control a cursor to track a continuously moving target under either rotated or mirror-reversed visual feedback. Although previous work has established that participants can learn to compensate for these perturbations during point-to-point movements, this compensation often seems to depend upon the use of re-aiming strategies – a solution that is time-consuming and therefore does not seem feasible in a task in which goals are constantly changing.

We found that both groups' tracking behavior was inconsistent with that of a re-aiming strategy, suggesting other mechanisms were used to compensate for these perturbations. The rotation group exhibited strong aftereffects once the perturbation was removed, amounting to an approximately

25° rotation of hand motion relative to target motion – consistent with previous findings in point-to-point tasks (*Taylor et al., 2010*; *Fernandez-Ruiz et al., 2011*; *Taylor and Ivry, 2011*; *Bond and Taylor, 2015*). This suggests that these participants learned to counter the rotation, at least in part, via adaptation. In contrast, participants who learned to compensate for the mirror-reversal showed no aftereffects, suggesting that they did not adapt their existing controller, but instead learned to compensate by establishing a *de novo* controller.

## The role of re-aiming strategies in *executing* tracking behavior

In principle, a target can be tracked by executing a series of intermittent catch-up movements. However, our results suggest that this possibility was unlikely for three reasons. First, under both perturbations, a majority of participants' tracking behavior could be accounted for by a linear model, and the additional variance in behavior that could be accounted for by a *nonlinear* model was comparatively small. This implies that participants tracked the target continuously, rather than intermittently, which would likely have introduced greater nonlinearities. Although it might be possible for very frequent catch-up movements to appear approximately linear, the frequency of such catch-up movements would have to be at least double the frequency of target motion being tracked (i.e., the Nyquist rate). The highest frequency at which participants were able to successfully compensate for the mirror reversal was around 1 Hz. This means participants would have had to generate at least two re-aimed movements per second to track the target smoothly at this frequency, a process that would have been fairly rapid and cognitively demanding over the course of a trial.

The second reason we reject the idea of repeated re-aiming is based on the delay between hand and target movement. Compensation for either of the perturbations introduced some additional tracking delay relative to baseline. However, this delay was less than 200 ms, which is smaller than would be expected if the participants had compensated by repeated strategic re-aiming. It has been demonstrated in some circumstances that re-aiming can occur in as little as 200 ms by caching the movement required for a given target location (*Huberdeau et al., 2019*; *McDougle and Taylor, 2019*). However, caching associations in this way appears to be limited to just two to seven discrete elements (*McDougle and Taylor, 2019*; *Collins and Frank, 2012*), and it seems doubtful that this mechanism could support a controller that must generate output when the state of the target (its location and velocity), as well as that of the hand, may vary in a continuous space.

Finally, participants' anecdotal reports also suggest they did not utilize a re-aiming strategy. After the experiment was complete, we asked participants to describe how they performed the tracking task under the perturbations. The vast majority of participants reported that when they tried to think about how to move their hand to counter the perturbations, they felt that their tracking performance deteriorated. Instead, they felt their performance was best when they let themselves respond naturally to the target without explicitly thinking about how to move their hands. Participants' disinclination to explicitly coordinate their hand movements provides further evidence against their use of a re-aiming strategy.

We believe, therefore, that it is unlikely that participants solved the tracking task under a mirror-reversal by using a deliberative re-aiming strategy that is qualitatively similar to that which has been described in the context of point-to-point reaching tasks. Instead, we believe that these participants constructed a new controller that instantiated a new, continuous mapping from current states and goals to actions.

However, it is possible that, given our experimental design, participants countered the perturbation in a way that is similar in some respects to traditional re-aiming and potentially indistinguishable from continuous control. Traditional accounts of re-aiming suggest that participants identify a fixed surrogate target location to aim their movements toward – effectively manipulating one of the inputs to the controller to achieve a particular desired output. Our results suggest that participants could not have performed the tracking task in this way. However, it is still possible for tracking to be performed by manipulating the input to a controller in a more general manner. For instance, the output of the tracking controller could depend on the instantaneous position *and* velocity of the target, and participants may have been able to counter the perturbation by manipulating these inputs to a fixed underlying controller in order to achieve output that would successfully track the target under the mirror reversal. Although this solution bears similarities to re-aiming, it differs significantly in that it entails modifying potentially many different inputs and doing so in a continuously changing manner.

Such a solution would be unlikely to be amenable to the deliberative processes responsible for static re-aiming and, in composite, could be considered a *de novo* controller.

## The role of re-aiming strategies in *acquiring* a *de novo* controller

Although our analyses revealed that participants did not primarily rely on an aiming strategy to *execute* continuous tracking movements, they could have initially depended on such a strategy to *acquire* the controller necessary to perform these movements. In a follow-up experiment, we tested whether limited practice in the point-to-point task would impair how well participants could learn to counter the rotation/mirror reversal. Although we found that both groups expressed less compensation for the perturbations compared to the main experiment, both groups still expressed some compensation, reproducing the qualitative features of learning from the main experiment. The fact that there are multiple explanations for this reduction in compensation (failure to develop a re-aiming strategy versus less time on task) makes it difficult to draw any strong conclusions from these results about what role re-aiming strategies play in acquiring a new controller.

However, previous evidence clearly demonstrates that people can learn to counter a mirror reversal using a re-aiming strategy when performing point-to-point reaches (*Wilterson and Taylor, 2019*). It is possible, therefore, that re-aiming strategies could contribute to acquiring a *de novo* controller. How exactly might such strategies contribute to learning? One possibility is that the deliberative computations performed when planning upcoming movements are used to help build a *de novo* controller. Alternatively, it may be easier for people to evaluate the quality of straight-line reaches (e.g., reach direction, movement time, task error) compared to tracking a pseudo-random trajectory, allowing them to update the parameters of a nascent controller more readily. Ultimately, the question of how a *de novo* controller is constructed is a major open question for future research.

## Frequency-domain signatures of adaptation and *de novo* learning

The pattern of compensation under the rotation and mirror-reversal was frequency specific (*Figure 5B*), with the nature of compensation at high frequencies revealing distinct signatures of adaptation and *de novo* learning between the two groups. At low frequencies, both groups of participants successfully compensated for their perturbations. But at high frequencies, only the rotation group was able to compensate; behavior for the mirror-reversal group at high frequencies was similar to baseline behavior. There were similarities, however, in the time course and frequency dependence of learning under each perturbation (*Figure 5B*), with both groups exhibiting a steady increase in compensation over time, particularly at lower frequencies. Additionally, both groups' compensation exhibited a similar diminution as a function of frequency.

We believe these results show that distinct learning processes drove two separate components of learning. One component, present only in the rotation group, was expressed uniformly at all frequencies and exhibited aftereffects, likely reflecting a parametric adjustment of an existing baseline controller, that is, adaptation. A second component of learning contributed to compensation in both groups of participants. This component was expressed primarily at low frequencies, exhibited a gradation as a function of frequency, and was not associated with aftereffects. We suggest this component corresponds to formation of a *de novo* controller for the task.

Although compensation for the rotation bore many hallmarks of adaptation, it also exhibited features of *de novo* learning seen in the mirror-reversal group, suggesting that participants in the rotation group employed a combination of the two learning processes. This is consistent with previous suggestions that residual learning under a visuomotor rotation that cannot be attributed to implicit adaptation may rely on the same mechanisms as those used for *de novo* learning (*Krakauer et al., 2019*). In summary, our data suggest that adaptation and *de novo* learning can be deployed in parallel to learn novel motor tasks.

## Potential control architectures supporting multiple components of learning

The properties of adaptation and *de novo* learning we have identified here can potentially be explained by the existence of two distinct control pathways, each capable of different forms of plasticity but with differing sensorimotor delays. An inability to compensate at high frequencies (when tracking an unpredictable stimulus; see *Roth et al., 2011*) suggests higher phase lags, potentially

due to greater sensorimotor delays or slower system dynamics; as phase lags approach the period of oscillation, it becomes impossible to exert precise control at that frequency. Therefore, we suggest that one control pathway may be slow but reconfigurable to implement arbitrary new controllers, while the other is fast but can only be recalibrated to a limited extent through adaptation.

It is possible that the two different control pathways that appear to learn differently might correspond to feedforward control (generating motor output based purely on target motion) and feedback control (generating motor output based on the cursor location and/or distance between cursor and target). Feedback control is slower than feedforward control due to the additional delays associated with observing the effects of one's earlier motor commands on the current cursor position. The observed pattern of behavior may thus be due to a fast but inflexible feedforward controller that responds rapidly to target motion, but always expresses baseline behavior (potentially recalibrated via implicit adaptation) interacting with a slow but reconfigurable feedback controller that responds to both target motion and the current cursor position. At low frequencies, the target may move slowly enough that any inappropriate feedforward control to track the target is masked by corrective feedback responses. But at high frequencies, the target may move too fast for feedback control to be exerted, leaving only inappropriate feedforward responses. It is not possible to dissociate the contributions of feedforward and feedback control on the basis of our current dataset, but in principle our approach can be extended to do so by including perturbations to the cursor position in addition to target movement (*Yamagami et al., 2019*; *Yamagami et al., 2020*).

An alternative possibility is that there may be multiple feedforward controllers (and/or feedback controllers) that incur different delays. A fast but inflexible baseline controller, amenable to recalibration through adaptation, might interact with a slower but more flexible controller. This organization parallels dual-process theories of learning and action selection (*Hardwick et al., 2019*; *Day and Lyon, 2000*; *Huberdeau et al., 2015*) and raises the possibility that the *de novo* learning exhibited by our participants might be, in some sense, cognitive in nature. Although we have rejected the possibility that participants countered the perturbation by repeated strategic re-aiming, recent theories have framed the prefrontal cortex as a general-purpose network capable of learning to perform arbitrary computations on its inputs (*Wang et al., 2018*). From this perspective, it does not seem infeasible that such a network could learn to implement an arbitrary continuous feedback controller that could compensate for the imposed perturbation or continuously modulate the input to an existing controller, albeit likely at the cost of incurring an additional delay over controllers that support task performance in baseline conditions.

## System identification as a tool for characterizing motor learning

Our characterization of learning made use of frequency-based system identification, a powerful tool that has been previously used to study biological motor control such as insect flight (*Fuller et al., 2014*; *Sponberg et al., 2015*; *Roth et al., 2016*), electric fish refuge tracking (*Cowan and Fortune, 2007*; *Madhav et al., 2013*), human posture (*Oie et al., 2002*; *Kiemel et al., 2006*), and human manual tracking (*Yamagami et al., 2019*; *Zimmet et al., 2020*). System identification and other sinusoidal perturbation techniques have previously been applied to characterize the trial-by-trial dynamics of learning from errors in adaptation tasks (*Baddeley et al., 2003*; *Ueyama, 2017*; *Miyamoto et al., 2020*). Our approach differs critically from these previous applications in that we use system identification to assess the state of learning and properties of the learned controller at a given time. In this latter sense, frequency-based system identification has not, to our knowledge, previously been applied to investigate motor learning. We have shown that this approach provides a powerful means to identify distinct forms of learning based on dissociable properties of the controllers they give rise to.

Our system identification approach has several advantages over other methods for studying motor control. In terms of practicality, this approach is more time efficient for data collection compared to the standard point-to-point reaches used in motor learning studies. Compared to time-domain methods, the frequency domain is particularly amenable for system identification given the rich suite of tools that have been developed for it (*Schoukens et al., 2004*). Moreover, our approach is also general as it can be applied to assess learning of arbitrary linear visuomotor mappings (e.g., 15˚ rotation, body-machine interfaces *Mussa-Ivaldi et al., 2011*). Under previous approaches, characterizing the quality of movements under different types of learned mappings (rotation, mirror-reversal) has necessitated different ad hoc analyses that cannot be directly compared (*Telgen et al.,*

*2014*). In contrast, our frequency-based approach provides a general method to characterize behavior under rotations, mirror-reversals, or any linear mapping from effectors to a cursor, owing to our 'multi-input multi-output' approach of identifying the $2 \times 2$ transformation matrix relating target movement and hand movement.

While the system identification approach used in the present study does capture learning, the results obtained using this approach do warrant careful interpretation. In particular, one must not interpret the empirical relationship that we measure between the target and hand as equivalent to the input-output relationship of the brain's motor controller. The former measures the response of the entire sensorimotor system to external input. The latter only measures how the controller sends motor commands to the body in response to input from the environment/internal feedback. Estimating the latter relationship requires a more nuanced approach that takes into account the closed-loop topology (*Roth et al., 2014*; *Yamagami et al., 2019*). Despite this, changes to the controller are still revealed using our approach; assuming that learning only drives changes in the input-output relationship of the controller – as opposed to, for example, the plant or the visual system – any changes in the overall target–hand relationship will reflect changes to the controller. Thus, our approach is a valid way to investigate learning.

Although the primary goal of our frequency-based analysis was to establish how participants mapped target motion into hand motion, system identification yields more detailed information than this; in principle, it provides complete knowledge of a linear system in that knowing how the system responds to sinusoidal input at different frequencies enables one to predict how the system will respond to arbitrary inputs. These data can be used to formally compare different possible control system architectures (*Zimmet et al., 2020*) supporting learning, and we plan to explore this more detailed analysis in future work.

## Mechanisms and scope of *de novo* learning

We have used the term '*de novo* learning' to refer to any mechanism, aside from implicit adaptation and re-aiming, that leads to the creation of a new controller. We propose that *de novo* learning proceeds initially through explicit processes before becoming cached or automatized into a more procedural form. There are, however, a number of alternative mechanisms that could be engaged to establish a new controller. One proposal is that *de novo* learning occurs by simultaneously updating forward and inverse models by simple gradient descent (*Pierella et al., 2019*). Another possibility is that a new controller could be learned through reinforcement learning. In motor learning tasks, reinforcement has been demonstrated to engage a learning mechanism that is independent of implicit adaptation (*Izawa and Shadmehr, 2011*; *Cashaback et al., 2017*; *Holland et al., 2018*) potentially via basal-ganglia-dependent mechanisms (*Schultz et al., 1997*; *Hikosaka et al., 2002*). Such reinforcement could provide a basis for forming a new controller. Although prior work on motor learning has focused on simply learning the required direction for a point-to-point movement, theoretical frameworks for reinforcement learning have been extended to continuous time and space to learn continuous controllers for robotics (*Doya, 2000*; *Theodorou et al., 2010*; *Smart and Kaelbling, 2000*; *Todorov, 2009*), and such theories could be applicable to how people learned continuous control in our experiment. However, it is important to note that regardless of the exact mechanism by which *de novo* learning occurs, our central claims from the present study still hold.

Although we have described the mirror-reversal task as requiring *de novo* learning, we acknowledge that there are many types of learning which might be described as *de novo* learning that this task does not capture. For example, many skills, such as playing the cello, challenge one to learn how to *execute* new movement patterns that one has never executed before (*Costa, 2011*). This is not the case in the tracking task which only challenges one to *select* movements one already knows how to execute. Also, in many cases, one must learn to use information from new sensory modalities for control (*van Vugt and Ostry, 2018*; *Bach-y-Rita and W Kercel, 2003*), such as using auditory feedback to adjust one's finger positioning while playing the cello. Our task, by contrast, only uses very familiar visual cues. Nevertheless, we believe that learning a new controller that maps familiar sensory feedback to well-practiced actions in a novel way is a critical element of many real-world learning tasks (e.g., driving a car, playing video games) and should be considered a fundamental aspect of any *de novo* learning.

Ultimately, our goal is to understand real-world skill learning. We believe that studying learning in continuous tracking tasks is important to bring us closer to this goal since a critical component of

many skills is the ability to continuously control an effector in response to ongoing external events, like in juggling or riding a bicycle. Studies of *well-practiced* human behavior in continuous control tasks has a long history, such as those examining the dynamics of pilot and vehicle interactions (*McRuer and Jex, 1967*). However, most existing paradigms for studying motor *learning* have examined only point-to-point movements. We believe the tracking task presented here offers a simple but powerful approach for characterizing continuous control and, as such, provides an important new direction for advancing our understanding of how real-world skills are acquired.

## Materials and methods

### Participants

Forty right-handed, healthy participants over 18 years of age were recruited for this study (24.28 ± 5.06 years old; 19 male, 21 female): 20 for the main experiment (*Figures 2–5*) and 20 for the follow-up experiment (*Figure 6*). Participants all reported having no history of neurological disorders. All methods were approved by the Johns Hopkins School of Medicine Institutional Review Board.

### Experimental tasks

Participants made planar movements with their right arm, which was supported by a frictionless air sled on a table, to control a cursor on an LCD monitor (60 Hz). Participants viewed the cursor on a horizontal mirror which reflected the monitor (*Figure 1B*). Hand movement was monitored at 130 Hz using a Flock of Birds magnetic tracker (Ascension Technology, VT) positioned near the participants' index finger. The (positive) $x$ axis was defined as rightward and the $y$ axis, forward. The cursor was controlled under three different hand-to-cursor mappings: (1) veridical, (2) 90° clockwise visuomotor rotation, and (3) mirror reversal about the 45° oblique axis in the $(x, y) = (1, 1)$ direction. Participants were divided evenly into two groups: one that experienced the visuomotor rotation ($n = 10$; four male, six female) and one that experienced the mirror reversal ($n = 10$; six male, four female). Both groups were exposed to the perturbed cursors while performing two different tasks: (1) the point-to-point task and (2) the tracking task. Each participant completed the experiment in a single session in 1 day.

#### Point-to-point task

To start a trial, participants were required to move their cursor (circle of radius 2.5 mm) into a target (gray circle of radius 10 mm) that appeared in the center of the screen. After 500 ms, the target appeared 12 cm away from the starting location in a random direction. Participants were instructed to move in a straight line, as quickly and accurately as possible to the new target. Once the cursor remained stationary (speed < 0.065 m/s) in the new target for 1 s, the target appeared in a new location 12 cm away, but constrained to lie within a 20 × 20 cm workspace. Different random target locations were used for each block. Blocks in the main experiment consisted of 150 reaches while blocks in the follow-up experiment (*Figure 6*) consisted of 15 reaches. To encourage participants to move quickly to each target, we provided feedback at the end of each trial about the peak velocity they attained during their reaches, giving positive feedback (a pleasant tone and the target turning yellow) if the peak velocity exceeded roughly 0.39 m/s and negative feedback (no tone and the target turning blue) if the peak velocity was below that threshold.

#### Tracking task

At the start of each trial, a motionless target (gray circle of radius 8 mm) appeared in the center of the screen, and the trial was initiated when the participant's cursor (circle of radius 2.5 mm) was stationary (speed < 0.065 m/s) in the target. From then, the target began to move for 46 s in a continuous, pseudo-random trajectory. The first 5 s was a ramp period where the amplitude of the cursor increased linearly from 0 to its full value, and for the remaining 41 s, the target moved at full amplitude. The target moved in a two-dimensional, sum-of-sinusoids trajectory where the movement in each axis was parameterized by amplitude, $\vec{a}$, frequency, $\vec{\omega}$, and phase, $\vec{\phi}$, vectors. The target's position, $r$, along one axis at time, $t$, was computed as

$$r = \sum_{i=1}^{7} a_i \cos(2\pi t\omega_i + \phi_i). \tag{1}$$

For the $x$-axis, $\vec{a} = [2.31, 2.31, 2.31, 1.76, 1.30, 0.97, 0.73]$ (cm) and $\vec{\omega} = [0.1, 0.25, 0.55, 0.85, 1.15, 1.55, 2.05]$ (Hz). For the $y$-axis, $\vec{a} = [2.31, 2.31, 2.31, 1.58, 1.03, 0.81, 0.70]$ (cm) and $\vec{\omega} = [0.15, 0.35, 0.65, 0.95, 1.45, 1.85, 2.15]$ (Hz). The elements of $\vec{\phi}$ were randomized for different blocks of the experiment, taking on values between $[-\pi, \pi]$. The amplitudes of the sinusoids for all but the lowest frequencies were proportional to the inverse of their frequency to ensure that each individual sinusoid had similar peak velocity. We set a ceiling amplitude for low frequencies in order to prevent target movements that were too large for participants to comfortably track.

Different frequencies were used for the $x$- and $y$-axes so that hand movements at a given frequency could be attributed to either $x$- or $y$-axis target movements. All frequencies were prime multiples of 0.05 Hz to ensure that the harmonics of any target frequency would not overlap with any other target frequency. The prime multiple design of the input signal ensured that there were no low-order harmonic relations between any of the component sinusoids on the input, making it likely that nonlinearities in the tracking dynamics would manifest as easily discernible harmonic interactions (i.e. extraneous peaks in the output spectra). Moreover, by designing discrete Fourier transform windows that were integer multiples of the base period (20 s, i.e., the inverse of the base frequency), any nonlinearities produced by taking the Fourier transform of non-periodic signals (i.e., non-integer multiples) were eliminated.

Participants were instructed to keep their cursor inside the target for as long as possible during the trial. The target's color changed to yellow any time the cursor was inside the target to provide feedback for their success. For the main experiment, one block of the tracking task consisted of eight, 46 s trials, while for the main experiment, one block consisted of six 46 s trials. Within each block, the same target trajectory was used for every trial. For different blocks, we randomized the phases of the target sinusoids to produce different trajectories. This produced five different target trajectories for participants to track in the six tracking blocks. The trajectory used for baseline and post-learning were the same to allow a better comparison for aftereffects. All participants tracked the same five target trajectories, but the order in which they experienced these trajectories was randomized in order to minimize any phase-dependent learning effects.

## Block structure

In the main experiment, we first assessed the baseline control of the rotation and mirror-reversal groups by having them perform one block of the tracking task followed by one block of the point-to-point task under veridical cursor feedback. We then applied either the visuomotor rotation or mirror reversal to the cursor and used the tracking task to measure their control capabilities during early learning. Afterwards, we alternated three times between blocks of point-to-point training and blocks of tracking. In total, each participant practiced their respective perturbation with 450 point-to-point reaches in between the early and late-learning tracking blocks. Finally, we measured aftereffects in the tracking task by removing the rotation/mirror reversal.

The follow-up experiment followed a similar block structure as the main experiment, but there were two differences of note. First, the number of point-to-point reaches was dramatically reduced per block to 15 reaches. Second, the number of point-to-point blocks was also reduced to 3 (one point-to-point block after the baseline, early, and late-learning tracking blocks), providing participants only 15 point-to-point reaches between the early and late-learning tracking blocks.

## Software

All non-statistical analyses were performed in MATLAB R2018b (The Mathworks, Natick, MA). All statistical analyses were performed in R version 4.0.2 (RStudio, Inc, Boston, MA) using the nlme and lsmeans packages (*R Development Core Team, 2016*; *Pinheiro et al., 2016*; *Lenth, 2016*). Figures were created using Adobe Illustrator (Adobe Inc, San Jose, CA).

## Point-to-point and trajectory-alignment analyses

In the point-to-point task, we assessed performance by calculating the angular error between the cursor's initial movement direction and the target direction relative to the start position. To

determine the cursor's initial movement direction, we computed the direction of the cursor's instantaneous velocity vector ~150 ms after the time of movement initiation. Movement initiation was defined as the time when the cursor left the start circle on a given trial.

In the tracking task, we assessed performance by measuring the average mean-squared error between the hand and target positions for every trial. For the alignment matrix analysis, we fit a matrix, $\hat{M} = \begin{bmatrix} a & b \\ c & d \end{bmatrix}$, that minimized the mean-squared error between the hand and target trajectories for every trial. In the latter analysis, the mean-squared error was additionally minimized in time by delaying the target trajectory relative to the hand. (While the time-delay allowed for the fairest possible comparison between the hand and target trajectories in subsequent analysis, changing or eliminating the alignment *did not* qualitatively change our results.) We estimated $\hat{M}$ as:

$$\hat{M} = \underset{M}{\operatorname{argmin}}\left\{ \begin{bmatrix} H_x \\ H_y \end{bmatrix} - M \begin{bmatrix} T_x \\ T_y \end{bmatrix} \right\} \tag{2}$$

where $H$ and $T$ represent hand and target trajectories. These estimated $\hat{M}$'s were averaged element-wise across participants to generate the alignment matrices shown in *Figure 3A*. These matrices were visualized by plotting their column vectors, also shown in *Figure 3A*.

The off-diagonal elements of each participant's alignment matrix were used to calculate the off-diagonal scaling, $S$, in *Figure 3B*:

$$S_{\text{rotation}} = \frac{-b+c}{2}, \qquad S_{\text{mirror}} = \frac{b+c}{2}. \tag{3}$$

Compensation angles, $\theta$, for the rotation group's alignment matrices were found using the singular value decomposition, $\operatorname{SVD}(\cdot)$. This is a standard approach which, as described in *Umeyama, 1991*, identifies a 2D rotation matrix, $R$, that best describes $\hat{M}$ irrespective of other transformations (e.g., dilation, shear) (*Figure 3C*, left). Briefly,

$$U\Sigma V^T = \operatorname{SVD}(\hat{M}^T), \tag{4}$$

$$R = VU^T \tag{5}$$

where $U$ and $V$ contain the left and right singular vectors and $\Sigma$ contains the singular values. Note that $R$ is a rotation matrix only if $\det(\hat{M}^T) \geq 0$, but $R$ is a reflection matrix when $\det(\hat{M}^T) < 0$. Although *Umeyama, 1991* have described a method whereby all $R$ can be forced to be a rotation matrix, we did not want to impose nonexistent structure onto $R$ and, thus, did not analyze trials which yielded reflection matrices. However, this was not a major issue for the analysis as nearly all trials yielded rotation matrices (3205 of 3360 data points for experiment 1; 2230 of 2520 data points for experiment 2). Subsequently, $\theta$ was calculated as

$$\theta = \operatorname{atan2}(R_{2,1}, R_{1,1}) \tag{6}$$

where $\operatorname{atan2}(\cdot)$ is the 2-argument arctangent and the inputs to the arctangent are elements of $R$ subscripted by the row and column numbers of the matrix.

Finally, for the mirror-reversal group, the scaling orthogonal to the mirror axis was found by computing how the matrix transformed the unit vector along the orthogonal axis (*Figure 3C*, right):

$$S_{\text{orthogonal}} = \frac{1}{2}\left( \begin{bmatrix} 1 & -1 \end{bmatrix} \begin{bmatrix} a & b \\ c & d \end{bmatrix} \begin{bmatrix} 1 \\ -1 \end{bmatrix} \right) = \frac{1}{2}(a - b - c + d). \tag{7}$$

## Frequency-domain analysis

To analyze trajectories in the frequency domain, we applied the discrete Fourier transform to the target and hand trajectories in every tracking trial. This produced a set of complex numbers representing the amplitude and phase of the signal at every frequency. We only analyzed the first 40 s of the trajectory that followed the 5 s ramp period so that our analysis period was equivalent to an integer multiple of the base period (20 s). This ensured that we would obtain clean estimates of the

sinusoids at each target frequency. Amplitude spectra were generated by taking double the modulus of the Fourier-transformed hand trajectories at positive frequencies.

The spectral coherence between signals was computed using Welch's periodogram technique, implemented using the MATLAB function mscohere. Windowing was performed using a 1040-sample Blackman–Harris window with 50% overlap between windows. To evaluate the proportion of participants' behavior that could be explained by a linear model, for every trial, we evaluated the single-input multi-output coherence at every frequency of target motion ('linear coherence'), determining how target motion in one axis elicited hand movement in both axes. This best captured the linearity of participants' behavior as using hand movement in only one axis for the analysis would only partially capture participants' responses to target movement at a given frequency. To evaluate the additional proportion of participants' behavior that could be explained by a nonlinear model but not a linear model, we computed the square root of the single-input single-output coherence (i.e., movements from the same axis) between hand movements from every pairwise combination of trials within each block ('nonlinear coherence'). Because this nonlinear coherence is calculated from data across trials, it cannot be computed on a trial-by-trial basis so we averaged this coherence within blocks to obtain one coherence measure per block. We then averaged the linear coherence within blocks and subtracted the linear coherence from the nonlinear coherence.

During each 40 s stimulus period, we assumed the relationship between target position and hand position behavior was well approximated by linear, time-invariant dynamics; this assumption was tested using the coherence analysis described above. Under this assumption, pure sinusoidal target motion at each frequency should be translated into pure sinusoidal hand motion at the same frequency but with different magnitude and phase. The relationship between hand and target can therefore be described in terms of a $2 \times 2$ matrix of transfer functions describing the behavior of the system at each possible frequency:

$$\begin{bmatrix} H_x(\omega) \\ H_y(\omega) \end{bmatrix} = P(\omega) \begin{bmatrix} T_x(\omega) \\ T_y(\omega) \end{bmatrix}, \quad P(\omega) = \begin{bmatrix} p_{xx}(\omega) & p_{xy}(\omega) \\ p_{yx}(\omega) & p_{yy}(\omega) \end{bmatrix}. \tag{8}$$

Here, $H(\omega)$ and $T(\omega)$ are the Fourier transforms of the time-domain hand and target trajectories, respectively, and $\omega$ is the frequency of movement. Each element of $P(\omega)$ represents a transfer function relating a particular axis of target motion to a particular axis of hand motion; the first and second subscripts represent the hand- and target-movement axes, respectively. Each such transfer function is a complex-valued function of frequency, which can further be decomposed into gain and phase components, for example:

$$p_{xy}(\omega) = g_{xy}(\omega) e^{j\phi_{xy}(\omega)}, \tag{9}$$

where $j$ is the imaginary number, $g_{xy}(\omega)$ describes the gain (ratio of amplitudes) between $y$-axis target and $x$-axis hand motion as a function of frequency, and $\phi_{xy}(\omega)$ describes the corresponding difference in the phase of oscillation.

We used this phase (in radians) to obtain the frequency-dependent lag between hand and target movement, $\delta(\omega)$, (in seconds) follows:

$$\delta(\omega) = \frac{\phi(\omega)}{2\pi\omega}. \tag{10}$$

The difference in $\delta(\omega)$ between baseline and late learning was used to generate *Figure 4C*.

We estimated the elements of $P(\omega)$ for frequencies at which the target moved by first noting that, for $x$-axis frequencies $\omega$, $T_y(\omega) = 0$. Consequently,

$$\begin{bmatrix} H_x(\omega) \\ H_y(\omega) \end{bmatrix} = \begin{bmatrix} p_{xx}(\omega) T_x(\omega) \\ p_{yx}(\omega) T_x(\omega) \end{bmatrix}, \tag{11}$$

and we can therefore estimate $p_{xx}(\omega)$ and $p_{yx}(\omega)$ as:

$$p_{xx}(\omega) = \frac{H_x(\omega)}{T_x(\omega)}, \quad p_{yx}(\omega) = \frac{H_y(\omega)}{T_x(\omega)}. \tag{12}$$

We estimated $p_{yx}(\omega)$ and $p_{yy}(\omega)$ analogously at $y$-frequencies of target motion.

These estimates yielded two elements of the overall transformation matrix $P(\omega)$ at each frequency of target movement. In order to construct a full $2 \times 2$ matrix, we paired the gains from neighboring $x$- and $y$-frequencies, assuming that participants' behavior would be approximately the same at neighboring frequencies. The resulting seven frequency pairings were ($x$ then $y$ frequencies reported in each parentheses in Hz): (0.1, 0.15), (0.25, 0.35), (0.55, 0.65), (0.85, 0.95), (1.15, 1.45), (1.55, 1.85), (2.05, 2.15).

The spatial transformation of target motion into hand motion at each frequency is described by the gain of each element of $P(\omega)$. However, gain and phase data can lead to certain ambiguities; for example, a positive gain with a phase of $\pi$ radians is indistinguishable from a negative gain with a phase of 0. Conventionally, this is resolved by assuming that gain is positive. In our task, however, the sign of the gain was crucial to disambiguate the directionality of the hand responses (e.g., whether the hand moved left or right in response to upward target motion). We used phase information to disambiguate positive from negative gains. Specifically, we assumed that the phase lag of the hand response at a given frequency would be the same across both axes of hand movement and throughout the experiment, but the gain would vary:

$$p_{xx}(\omega) \approx g_{xx}(\omega)e^{j\widetilde{\phi}(\omega)}, \qquad p_{yx}(\omega) \approx g_{yx}(\omega)e^{j\widetilde{\phi}(\omega)}. \tag{13}$$

For a given movement frequency, $\widetilde{\phi}(\omega)$ was set to be the same as the mean phase lag during the baseline block, where the gain was unambiguously positive. This assumption enabled us to compute a signed gain for each transfer function by taking the dot product between the transfer function and $e^{j\widetilde{\phi}(\omega)}$. This method thus yielded gains for each axis of hand motion, at each target frequency, and at each point during learning.

As we did for the transfer-function matrix $P(\omega)$, we paired the gains from neighboring frequencies to obtain a series of seven gain matrices which geometrically described how target motion was translated into hand motion from low to high frequencies. Similar to the alignment matrix analysis, visualizations of these gain matrices were constructed by plotting the column vectors of the matrices. Off-diagonal gain, rotation angle, and gain orthogonal to the mirroring axis were calculated in the same way as in *Equations 2–6*.

## Statistics

The primary statistical tests for the main and follow-up experiments were performed using linear mixed-effects models. These models were fit using data from three parts of the study: (1) alignment matrix analysis in the main experiment, (2) gain matrix analysis in the main experiment, and (3) gain matrix analysis in the follow-up experiment. The data used in these models were the off-diagonal values of the transformation and gain matrices. In all models, data from the first trial of baseline, the last trial of late learning, and the first trial of post-learning were analyzed. No outlier rejection was performed for these analyses. Using Wilkinson notation, the structure of the model for the alignment matrix analysis was [off-diagonal scaling] ~ [block of learning] * [perturbation group], while the structure for both gain matrix analyses was [off-diagonal gain] ~ [block of learning] * [perturbation group] * [frequency of movement]. Data were grouped within subjects (subjects were considered a random effect of the model).

We subsequently performed post hoc statistical comparisons as needed for each of the linear mixed-effects models. For the alignment matrix analysis, we performed pairwise comparisons using Tukey's range test. For the gain matrix analysis in the main and follow-up experiments, there was a three-way interaction between frequency and the other regressors, so we fit seven different mixed-effects models for each frequency of movement post hoc. We performed pairwise comparisons on these frequency-specific models using Tukey's range test. Although this test corrects for multiple comparisons, it only corrected the p-values for comparisons within each of the seven frequency-specific models. Because we ran Tukey's range test seven times in total, we applied an additional Bonferroni correction by multiplying the p-values by seven.

We used a two-way ANOVA to compare the late-learning gain matrices between the main and follow-up experiments. Similar to the linear mixed-effects analyses, we compared the off-diagonal elements of the matrices. No outlier rejection was performed for this analysis. Using Wilkinson

notation, the structure of the ANOVA was [off-diagonal gain] ~ [experiment] * [perturbation group] * [frequency of movement].

## Acknowledgements

We thank Amanda Zimmet, John Krakauer, Amy Bastian, and Christopher Fetsch for immensely helpful discussions. This material is based upon work supported by the National Science Foundation under Grant No. 1825489. CSY was supported by NIH 5T32NS091018-17, 5T32NS091018-18, and the Link Foundation Modeling, Simulation and Training Fellowship.

## Additional information

### Funding

| Funder | Grant reference number | Author |
|---|---|---|
| Link Foundation | Modeling, Simulation and Training Fellowship | Christopher S Yang |
| National Institutes of Health | 5T32NS091018-17 | Christopher S Yang |
| National Institutes of Health | 5T32NS091018-18 | Christopher S Yang |
| National Science Foundation | 1825489 | Noah J Cowan |

The funders had no role in study design, data collection and interpretation, or the decision to submit the work for publication.

### Author contributions

Christopher S Yang, Conceptualization, Data curation, Software, Formal analysis, Funding acquisition, Validation, Investigation, Methodology, Writing - original draft, Writing - review and editing; Noah J Cowan, Conceptualization, Formal analysis, Supervision, Funding acquisition, Visualization, Methodology, Writing - original draft, Writing - review and editing; Adrian M Haith, Conceptualization, Formal analysis, Supervision, Visualization, Methodology, Writing - original draft, Project administration, Writing - review and editing

### Author ORCIDs

Christopher S Yang https://orcid.org/0000-0002-7645-3861
Noah J Cowan https://orcid.org/0000-0003-2502-3770
Adrian M Haith https://orcid.org/0000-0002-5658-8654

### Ethics

Human subjects: Informed consent and consent to publish was obtained from all participants included in this work. All methods were approved by the Johns Hopkins School of Medicine Institutional Review Board under NA_00048918.

### Decision letter and Author response

Decision letter https://doi.org/10.7554/eLife.62578.sa1
Author response https://doi.org/10.7554/eLife.62578.sa2

## Additional files

### Supplementary files

• Transparent reporting form

### Data availability

The data and code used to produce the results in this study can be found in the Johns Hopkins University Data Archive (https://doi.org/10.7281/T1/87PH8T).

The following dataset was generated:

| Author(s) | Year | Dataset title | Dataset URL | Database and Identifier |
|---|---|---|---|---|
| Yang CS, Cowan NJ, Haith AM | 2021 | Data and software associated with the publication "De novo learning versus adaptation of continuous control in a manual tracking task" | https://doi.org/10.7281/T1/87PH8T | Johns Hopkins University Data Archive, 10.7281/T1/87PH8T |

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
