## [Decision Letter]

**Acceptance summary:**

All three reviewers (including the Reviewing Editor) were impressed with the changes to the manuscript. They felt that the revised work strongly justifies the conclusion that humans can rapidly and flexibly shift control policies in response to environmental perturbations.

**Decision letter after peer review:**

Thank you for submitting your article "*De novo* learning and adaptation of continuous control in a manual tracking task" for consideration by *eLife*. Your article has been reviewed by 3 peer reviewers, including Timothy Verstynen as the Reviewing Editor and Reviewer #1, and the evaluation has been overseen by Tamar Makin as the Senior Editor.

The reviewers have discussed the reviews with one another and the Reviewing Editor has drafted this decision to help you prepare a revised submission.

Summary:

This work looks at "*de novo* learning" in the context of fast continuous tasks, i.e., shifts of control policies (or controllers), rather than parameter changes in existing policies, with visuomotor adaptation. In a set of 2 experiments, using a mixture of discrete point-to-point movement trials and continuous tracking of moving target trials, the authors set out to determine whether the structure of shifts between visual and proprioceptive information determines whether learning relies on adaptation or shifts in control policies. Using both the presence of post-shift aftereffects and trailwise model fitting, the authors find that, simple rotations of visual inputs of the hand lead primarily to changes in control parameters while mirror reversals lead to changes in the control policy itself. Although there was evidence for a mixture of adaptation and *de novo* learning in both conditions. The authors infer from this evidence that humans can rapidly and flexibly shift control policies in response to environmental perturbations.

In general, this was a very cleverly designed and executed set of studies. The theoretical framing and experimental design are clean and clear. The data is compelling on the existence of condition differences. However, all three reviewers identified significant concerns that should be addressed.

Essential revisions:

1. Inferential logic

Reviewer #1 pointed out that there are two key parts to the analyses used to infer that mirror-rotations lead to *de novo* policy shifts while rotations lead to adaptation. The first is the presence of post-perturbation aftereffects. While we clearly see stronger aftereffects in the rotation condition than in the mirror reversal condition, suggesting a difference in fundamental control mechanisms, it is not clear why control policy shifts are the only alternative explanation for attenuated aftereffects. I'm pretty sure that this is just a confusion based on how the problem is posed in the paper.

The second are the alignment matrices (in both immediate hand position and movement frequency spaces), that are estimated based on model fits to the data. I'll consider both in turn. Perhaps more problematically, the alignment matrices (Figure 3A) and vectors (Figure 3A, 5B, 6B), based on the model fits, show a very high degree of variability across conditions and do not perfectly align to the simple predictions shown in Figure 3A. While the reviewer agrees that if you squint on the mean vector direction, they look qualitatively consistent with the models, but only qualitatively. In fact, the fits to the "ideal" shifts or rotations (Figure 5C, 6C) suggest only partial alignment to the pure models. How are we sure that this isn't reflecting an alternative mechanism, instead of partial *de novo* learning?

In both the aftereffect and alignment fit analyses, the inference for *de novo* learning seems to be based on either a null (i.e., no aftereffect in mirror-rotation) or partial fits to a specific model. This leaves the main conclusions on somewhat shaky ground.

Reviewer #2 raised similar concerns, pointing out that the authors introduce the concept of *de novo* learning in contrast to both error-driven adaptation and re-aiming: 'a motor task could be learned by forming a *de novo* controller, rather than through adaptation or re-aiming.' However, the discussion reframes *de novo* learning as purely in contrast with implicit adaptation: '[…] *de novo* learning refers to any mechanism, aside from implicit adaptation, that leads to the creation of a new controller'. While this apparent shift in perspective is likely due to their results and realistically represents the scientific process, this shift should be more explicitly communicated.

As explicitly raised in the discussion and suggested in the introduction, the authors have categorized any learning process that is not implicit adaptation as a *de novo* learning process. To substantiate this conceptual decision, the authors should further explain why motor learning unaccounted for by established learning processes should be accounted for by a *de novo* learning process.

This same reviewer also pointed out that, participants could not learn mirror-reversal under continuous tracking without the point-to-point task, which the authors interpret to mean that re-aiming is important for the ‘acquisition’ of a *de novo* controller. This suggests that re-aiming may not be important for the ‘execution’ of a *de novo* controller.

However, the frequency-based performance analysis presented in the main experiment would seem to suggest otherwise. As mentioned in the introduction, low stimulus frequencies allow a catch-up strategy. Both rotation and mirror groups were successful at compensating at low frequencies, but the mirror-reversal group was largely unsuccessful at high frequencies. Assuming that higher frequencies inhibit cognitive strategy, this suggests to me that catch-up strategies might be essential to mirror-reversal, possibly not only during learning but also during execution.

Further, the authors note that, in the rotation group, aftereffects only accounted for a fraction of total compensation, then suggest that residual learning not accounted for by adaptation was attributable to the same *de novo* learning process driving mirror reversal. This framing makes it unclear to me how the authors think re-aiming fits into the concept of a *de novo* learning process (e.g. Is all learning not driven by implicit adaptation *de novo* learning? What about the role of re-aiming?)

Reviewer #3 points out that in the abstract, the last line says, 'Our results demonstrate that people can rapidly build a new continuous controller *de novo* and can flexibly integrate this process with adaptation of an existing controller'. It's not clear if the authors have shown the latter definitively. What is the reasoning for this statement, "flexibly integrate this process with adaptation of an existing controller"? It would seem you would need the same subjects to perform both experimental tasks (mirror reversal and VMR) concurrently to make this claim.

Reviewer #3 also points out that, on lines 339-342, the results show that mirror-reversal learning is low at high frequencies (Figure 5B). The authors interpret this as reason to believe that this is actually de-novo learning and not adaptation of an existing controller. This seems somewhat unfounded. Could it be that *de novo* learning performs well at low frequency, through 'catch-up' movements, but not at high frequencies? Do the authors have a counter argument for this explanation?

On lines 343 – 350, Reviewer #3 points out that the authors ascribe the difference between after-effects and end of learning to be due to de-novo learning even in the rotation group. However, that difference would likely be due to the use of explicit strategy during learning and its disengagement afterwards, or perhaps a temporally labile learning. Can the authors rule these possibilities out? What were the instructions given at the end of the block and how much time elapsed?

2. Linearity analysis

Reviewer #1 reported having a hard time understanding the analysis leading to the conclusion that there is a linear relationship between target motion and hand motion. The logic of the spectral analysis was not clear to me, and the results shown in Figure 4 were not intuitive. In addition, there was no actual quantification used to make a conclusion about linearity. Thus, it was difficult to determine whether this aspect of the authors' conclusion (a critical inference for them to justify their main conclusion) was correct.

Reviewer #2 raised similar concerns, pointing out that using linearity as a metric for mechanistic inference has limitations.

– The absence of learning (errors) would present as nonlinearity.

– The use of cognitive strategy could present as nonlinearity.

– It doesn't seem possible to parse the two mechanisms, especially as you might expect both an increase in error at the beginning of learning and possibly an intervening cognitive strategy at the beginning of learning.

Given these issues, a more grounded interpretation is that linearity simply represents real-time updating. If the relationship between the cursor and the hand is nonlinear, then updating is not in real time.

The data shown in Figure 4B do not appear to provide clear evidence that the relationship between the cursor and the hand was approximately linear. Currently, it seems equally plausible to say that the data are approximately non-linear. Establishing a criterion for nonlinearity would be useful (e.g. shuffling a linear response for comparison). This was also pointed out by Reviewer #3 who pointed out that details about frequency analysis are buried deep in the methods (around line 711), especially how the hand-target coherence (shown in 4B) is calculated. It would be helpful to include some of these details in the main text. For example, it is currently very difficult to understand the relationship when from moving from Figure 4A to 4B.

Reviewer #3 raises a similar concern. The authors show the tracking strategies participants applied by investigating the relationship between hand and target movement. The linear relationship would suggest that participants tracked the target using continuous movements. In contrast, a nonlinear relationship would suggest that participants used an alternative tracking strategy. The authors only state this relationship is based on figure 4 but it seems do not provide any proof of the linearity. It would be more convincing to provide an analysis to show that the relationship is indeed linear or nonlinear.

3. Statistical results

Reviewer #1 points out that any of the key statistical results were buried in the main text and some were incompletely reported. Can the authors provide a table (or set of tables) of the key statistics, including at least the value of the statistical test itself and the p-value, if not also estimates of confidence on the estimates?

Reviewer #3 also points out that outlier rejection based on some subjects who had greatly magnified, or attenuated data seems like it might be biasing the data. Also, the outlier rejection criteria used (>1.5 IQR) seems very stringent. Furthermore, it appears there was no outlier rejection on the main experiment. It would be good to be consistent across experiments.

4. Experiment 2

The intention for experiment 2 is to see how much training on the point-to-point task influenced adaptation mechanisms during the tracking task. Yet, this experiment still included extensive exposure to the point-to-point task. Just not as much as in experiment 1. Given this, how can an inference be cleanly made about the influence of one task on the other? Wouldn't the clean way to ask this question be to just not run the point-to-point tracking task at all?

5. Frequency analysis

The authors state that "The failure to compensate at high frequencies.… is consistent with the observation that people who have learned to make point-to-point movements under mirror-reversed feedback are unable to generate appropriate rapid corrections to unexpected perturbations." This logic is not clear. How is this inferred based on which movement frequencies show an effect, and which do not, leading to this conclusion?

6. Clarity of logic

Reviewer #3 states that would be helpful if the authors could provide more background/context on their view of *de novo* learning and explanations on relationship between *de novo* learning and the adapted controller model. For example, why does the lack of aftereffects under the mirror-reversal imply that the participants did not counter this perturbation via adaptation and instead engaged the learning by forming a *de novo* controller (Line 199)? Is the reasoning purely behavioral observations, or is there a physiological basis for this assertion?

In addition, this same reviewer points out that on lines 197-199: The reason for the lack of after-effects in the mean-squared error analysis is a little vague. It took a few tries to understand the reasoning. It would be good to spell this out a little more clearly. In lines 223-225: The logic behind why coupling across axes is not nonlinear behavior seems to be missing. It's quite unclear and currently difficult to understand. It would be very helpful to spell this out too.

7. Learning in the visuomotor rotation (VMR) condition.

Reviewer #3 also shows that surprisingly, there is no measurement of aiming in the learning to VMR. Several motor learning studies (several the authors cite) show that learning in VMR is a combination of implicit and explicit. It is understood that this is not possible in the continuous tracking task, but can certainly be done in the point to point task. Is there a reason this was not done? Wouldn't this have further supported the author's claim of an existing controller?

---

## [Author Response]

Essential revisions:1. Inferential logicReviewer #1 pointed out that there are two key parts to the analyses used to infer that mirror-rotations lead to *de novo* policy shifts while rotations lead to adaptation. The first is the presence of post-perturbation aftereffects. While we clearly see stronger aftereffects in the rotation condition than in the mirror reversal condition, suggesting a difference in fundamental control mechanisms, it is not clear why control policy shifts are the only alternative explanation for attenuated aftereffects. I'm pretty sure that this is just a confusion based on how the problem is posed in the paper.

We thank the reviewer for this comment. We argue that there are three different possibilities for how the brain learns new motor tasks: (1) adaptation (parametrically changing the properties of an existing controller), (2) re-aiming (specifying an alternate movement goal to an existing controller so as to achieve a particular desired outcome), and (3) *de novo* learning (switching to an alternative controller which has been newly instantiated). We argue that neither adaptation nor re-aiming can explain participants’ behavior in the mirror reversal group (based on the lack of aftereffects and the linearity analysis in Figure 4). We therefore conclude that this group compensated for the mirror reversal by building a new controller.

Our reasoning that learning must either be achieved by leveraging an existing controller or instantiating a new one parallels that of Telgen and colleagues (2014)^1^, who first introduced the idea that mirror reversal might be learned by forming a new controller *de novo*. They, however, overlooked the possibility that learning to generate point-to-point movements under a mirror reversal might be accomplished by a simple re-aiming mechanism. Indeed, subsequent work has shown that this is likely to be the case^2^, calling into question the conclusion that participants are able to build a new controller in order to compensate for a mirror reversal.

We think that one likely source of confusion is that our use of the term “*de novo* learning” appears to describe a specific putative mechanism of learning. However, this is not what we mean; by the phrase “*de novo* learning,” we intend to describe any process by which a *de novo* controller might be created. We are agnostic as to the specific learning mechanisms (e.g. reinforcement learning, error-driven learning, automatization of cognitive strategies) that might bring this about.

We have revised much of the Introduction to clarify our logic and more concretely pose our conceptual framework to the reader. Additionally, we have addressed these issues in lines 461-547 of the Discussion.

The second are the alignment matrices (in both immediate hand position and movement frequency spaces), that are estimated based on model fits to the data. I'll consider both in turn. Perhaps more problematically, the alignment matrices (Figure 3A) and vectors (Figure 3A, 5B, 6B), based on the model fits, show a very high degree of variability across conditions and do not perfectly align to the simple predictions shown in Figure 3A. While the reviewer agrees that if you squint on the mean vector direction they look qualitatively consistent with the models, but only qualitatively. In fact, the fits to the "ideal" shifts or rotations (Figure 5C, 6C) suggest only partial alignment to the pure models. How are we sure that this isn't reflecting an alternative mechanism, instead of partial *de novo* learning?

We would like to clarify that the vectors illustrated in Figure 3A are not intended as predictions, but rather as a reference to help illustrate to the reader what ideal compensation for the mirror reversal should look like in this type of plot. The imperfect alignment between this reference and participants’ behavior does indeed show that compensation is only partial. However, we would not characterize this as “partial *de novo* learning”. Rather, the compensation that *did* occur was fully achieved by creating a new controller *de novo*, even though it did not achieve full compensation.

We also want to emphasize that the plotted vectors should not really be thought of as “model fits”. They are direct estimates of the (frequency-dependent) transformation relating target movement to hand movement and, in this respect, they are better thought of as more akin to initial movement direction for point-to-point reaches, rather than as model fits.

We have revised the manuscript to clarify the interpretation of these matrices in lines 161-163 and 167-171.

In both the aftereffect and alignment fit analyses, the inference for *de novo* learning seems to be based on either a null (i.e., no aftereffect in mirror-rotation) or partial fits to a specific model. This leaves the main conclusions on somewhat shaky ground.

As addressed in response 1.2, the reference in Figure 3A was never intended to represent a ‘model’ of *de novo* learning. The alignment/gain matrix analyses are a means of quantifying the behavior of participants and was never meant to provide a prediction under one hypothesis or another. So the inference is not based on a ‘partial fit to a specific model’. Instead, the inference is based on the differences in aftereffects across conditions.

Regarding the null result for the after-effects, our conclusion is not solely drawn based on the absence of an aftereffect but rather is based on the difference in aftereffects across conditions. We realized that we had neglected to report these statistical differences across groups in the original manuscript. We agree that if we had only tested the mirror-reversal group this would be a significant weakness, but the comparison to the rotation group shows a very clear difference in learning processes.

We have now included explicit comparisons of aftereffect size across groups in the Results in lines 197-200, 397-399, and 439-441.

Reviewer #2 raised similar concerns, pointing out that the authors introduce the concept of *de novo* learning in contrast to both error-driven adaptation and re-aiming: 'a motor task could be learned by forming a *de novo* controller, rather than through adaptation or re-aiming.' However, the discussion reframes *de novo* learning as purely in contrast with implicit adaptation: '[…] *de novo* learning refers to any mechanism, aside from implicit adaptation, that leads to the creation of a new controller'. While this apparent shift in perspective is likely due to their results and realistically represents the scientific process, this shift should be more explicitly communicated.

We thank the reviewer for pointing out this passage, which was poorly worded. This statement was not intended to mark a shift in the central claim of our paper, and it was an oversight not to explicitly mention re-aiming here. Re-aiming does not lead to the creation of a new controller; rather, it works by altering the movement goal that is fed to an existing controller. We argue that the mirror reversal group’s tracking behavior could not be explained by either adaptation or re-aiming. We have corrected this statement in lines 658-659 to maintain clearer and more consistent messaging throughout the paper.

As explicitly raised in the discussion and suggested in the introduction, the authors have categorized any learning process that is not implicit adaptation as a *de novo* learning process. To substantiate this conceptual decision, the authors should further explain why motor learning unaccounted for by established learning processes should be accounted for by a *de novo* learning process.

We thank the reviewer for this comment. We do define the term “*de novo* learning” broadly as any learning that does not proceed either by adaptation *or* by re-aiming. This follows the same framing as Telgen and colleagues (2014)^1^, except that we also consider re-aiming as a possible means of compensation.

We also think a potentially confusing aspect of our framing was a failure on our part to clearly distinguish the product of learning (a *de novo* controller) and the learning process itself (*de novo* learning). While the former is well defined, the latter is not. We do not make any qualitative claims about the learning process that brings about the instantiation of this new controller. We do believe, however, that our findings rule out both adaptation and re-aiming as potential learning processes. When we refer to ‘*de novo* learning processes,’ we intend to refer to whatever processes are responsible for learning a new controller (after having ruled out adaptation or re-aiming). At present, little is understood about how such learning might proceed. Our central claim is, however, agnostic to the exact mechanism by which a new controller is built, and our goal was not to characterize the learning mechanism itself but rather to characterize the properties of the learned behavior.

We have revised our Discussion (see section titled “Mechanisms and Scope of *de novo* Learning”) to better explain our reasoning on this point. We have also reframed the Introduction and key parts of the Results to better emphasize that our paper identifies the formation of a new controller (in contrast to adaptation of an existing controller or leveraging an existing controller by re-aiming) and avoid giving the impression that our experiments identify the process by which this new controller is established.

This same reviewer also pointed out that, participants could not learn mirror-reversal under continuous tracking without the point-to-point task, which the authors interpret to mean that re-aiming is important for the ‘acquisition’ of a *de novo* controller. This suggests that re-aiming may not be important for the ‘execution’ of a *de novo* controller.However, the frequency-based performance analysis presented in the main experiment would seem to suggest otherwise. As mentioned in the introduction, low stimulus frequencies allow a catch-up strategy. Both rotation and mirror groups were successful at compensating at low frequencies but the mirror-reversal group was largely unsuccessful at high frequencies. Assuming that higher frequencies inhibit cognitive strategy, this suggests to me that catch-up strategies might be essential to mirror-reversal, possibly not only during learning but also during execution.

We thank the reviewer for raising this very important point regarding whether the mirror-reversal group may be using a catch-up strategy. The fact that participants do not appropriately compensate for the mirror reversal at high frequencies may be because the time required to deploy a catch-up (i.e., re-aiming) strategy is too long to be effective at high frequencies (i.e., longer than the period of the sinusoids). Previously, we noted that the fast-paced tracking task would have precluded time-consuming re-aiming and appealed to the idea that participants behavior was approximately linear as evidence that they were not performing intermittent ‘catch-up’ movements. We agree with the reviewer, however, that this was not entirely rigorous. We realized that even if participants perform a series of discretely planned movements, their overall behavior might appear linear depending on how rapidly the movements could be planned.

We have now included an additional analysis (Figure 4C) which we believe provides more convincing evidence that participants did not solve the mirror-reversal through a series of discretely planned ‘catch-up’ movements. We examined the lag between hand and target movements to determine whether this lag provided enough time to apply a re-aiming strategy under the rotation/mirror reversal. Based on previous literature^3,4^, we expected that if one were to use a re-aiming strategy to counter a large rotation/mirror reversal, this would incur an additional ~300 ms of reaction time on top of that required at baseline. However, in measuring the increase in lag between baseline and late learning, we found that the lag only increased by an average of 83 and 191 ms for the rotation and mirror-reversal groups, respectively. Together with the coherence analysis, this suggests that participants did not employ a catch-up strategy. We have included the lag analysis in lines 279-297 in the Results.

Additionally, participants’ anecdotal reports also suggest they did not utilize a re-aiming strategy. After the experiment was complete, we asked participants to describe how they performed the tracking task under the perturbations. The vast majority of participants reported that when they tried to think about how to move their hand to counter the perturbations, they felt that their tracking performance deteriorated. Instead, they felt their performance was best when they let themselves respond naturally to the target without explicitly thinking about how to move their hands. Participants’ disinclination to explicitly coordinate their hand movements provides further evidence against their use of a re-aiming strategy. We have mentioned these informal reports in lines 500-507.

We do acknowledge that we cannot entirely rule out the possibility that participants used a re-aiming strategy. However, in order for re-aiming to be viable, it would have to be extremely rapid—more rapid than almost all accounts of re-aiming have suggested. Moreover, cases where participants have been found to apply re-aiming strategies extremely rapidly (of the order of ~200ms^4,5^) by “caching” the solution for each target tend to include only a small number of static targets with a fixed start position. This strategy does not appear to be scalable to cases where there are 12 or more possible targets^4^. In the tracking task, the possible state space of the target and hand is vastly greater than for center-out reaching tasks, making it unlikely that a re-aiming + caching solution could be feasible, at least as it is currently understood. We have revised the section in the Discussion titled “The Role of Re-aiming Strategies in Executing Tracking Behavior” to address these points.

Further, the authors note that, in the rotation group, aftereffects only accounted for a fraction of total compensation, then suggest that residual learning not accounted for by adaptation was attributable to the same *de novo* learning process driving mirror reversal. This framing makes it unclear to me how the authors think re-aiming fits into the concept of a *de novo* learning process (e.g. Is all learning not driven by implicit adaptation *de novo* learning? What about the role of re-aiming?)

As described in responses 1.1, 1.4, and 1.5, our original discussion may have been unclear about how we view the relationship between *de novo* learning and re-aiming. It perhaps incorrectly gave the impression that we believe learning is a dichotomy between implicit adaptation and *de novo* learning and that we believe re-aiming is a form of *de novo* learning. As described in responses 1.1, 1.4, and 1.5, we have revised the Introduction, Results, and Discussion to clarify our conceptualization of these learning processes and more carefully explain the differences between a *de novo* learned controller and use of a re-aiming strategy.

Reviewer #3 points out that in the abstract, the last line says, 'Our results demonstrate that people can rapidly build a new continuous controller *de novo* and can flexibly integrate this process with adaptation of an existing controller'. It's not clear if the authors have shown the latter definitively. What is the reasoning for this statement, "flexibly integrate this process with adaptation of an existing controller"? It would seem you would need the same subjects to perform both experimental tasks (mirror reversal and VMR) concurrently to make this claim.

We agree with the reviewer that our claim that *de novo* learning can “flexibly integrate” with adaptation was not as clear as it should be. Our intention was to point out that *de novo* learning and adaptation can operate in tandem to learn new tasks, as seen in the rotation group (explained in lines 343–350 and more extensively discussed in lines 424–445 of the initially submitted manuscript). The basis for this conclusion is that these participants expressed aftereffects, which suggested they engaged adaptation during learning, but the aftereffect’s magnitude was only a fraction of the total compensation exhibited at late learning. This suggested that some component of the compensation for the rotation was immediately disengaged after the rotation was removed, and we believe this component to be *de novo* learning. We have changed the words “flexibly integrate” in the Abstract to “simultaneously deploy” to communicate our conclusion more clearly. We have also added a sentence in the discussion to emphasize this point (lines 570-571).

Regarding the reviewer’s suggestion to have subjects perform both the mirror reversal and rotation simultaneously, this visuomotor perturbation simply amounts to a mirror reversal where the mirroring axis is oriented differently from that of the originally applied reversal. Thus, we would expect that this perturbation would only engage de novo learning and not adaptation.

Reviewer #3 also points out that, on lines 339-342, the results show that mirror-reversal learning is low at high frequencies (Figure 5B). The authors interpret this as reason to believe that this is actually de-novo learning and not adaptation of an existing controller. This seems somewhat unfounded. Could it be that *de novo* learning performs well at low frequency, through 'catch-up' movements, but not at high frequencies? Do the authors have a counter argument for this explanation?

We thank the reviewer for pointing out our claim here was unclear. We were not suggesting that the mirror reversal group’s lack of compensation at high frequencies is a reason to believe that they engaged *de novo* learning. Instead, we claim that this group engaged *de novo* learning because their behavior was inconsistent with adaptation (based on lack of aftereffects) and the use of a re-aiming strategy (based on analysis in Figure 4).

Compensating for the perturbation via frequent ‘catch-up’ movements would not be a form of *de novo* learning, but rather repeated re-aiming. The issue of whether the frequency-dependence of compensation under the mirror-reversal (and rotation) might reflect a series of ‘catch-up’ movements is an important one which the reviewers’ comments have prompted us to consider more deeply and provide better evidence against. We have provided a more extensive argument in response 1.6 detailing why we do not believe that participants’ performance in the mirror-reversal group is consistent with a series of catch-up movements.

To more clearly explicate our claim that the mirror-reversal group learned by creating a *de novo* controller, we have edited lines 408-413 to emphasize that this claim is based on the fact that this group’s behavior was inconsistent with adaptation and a re-aiming strategy.

On lines 343 – 350, Reviewer #3 points out that the authors ascribe the difference between after-effects and end of learning to be due to de-novo learning even in the rotation group. However, that difference would likely be due to the use of explicit strategy during learning and its disengagement afterwards, or perhaps a temporally labile learning. Can the authors rule these possibilities out? What were the instructions given at the end of the block and how much time elapsed?

Before the post-learning block, we verbally informed participants that the rotation/mirror reversal would be disengaged and that their cursor control would revert to normal.

The time elapsed between the last perturbation trial and the first aftereffect trial was not different from the time elapsed in between other blocks of perturbation trials (~30 seconds). Therefore, the lack of aftereffects cannot be attributed to temporal lability of the learned compensation.

As we discussed in detail in response 1.6, we do not believe that the difference between aftereffects and performance at the end of learning can be ascribed to use of an explicit strategy—at least not in a form that is qualitatively similar to what has been extensively described in the case of point-to-point movements. As we argued in the paper initially, explicit strategies are known to be time-consuming to implement, which would prohibit them from being used to solve a continuous tracking task like the one we used. We have strengthened this argument by analyzing in more detail the additional tracking delays introduced by having to track the perturbation (<200 ms) and comparing this to previous estimates of the time required for re-aiming (~300ms). Although there is some evidence that explicit strategies can become cached and, thereby, deployed much more rapidly, this appears to occur only in very limited scenarios with very few targets and does not seem possible when there are 12 or more targets, let alone a whole workspace (as there is in the tracking task). We have updated lines 279-297 to include this new analysis.

2. Linearity analysisReviewer #1 reported having a hard time understanding the analysis leading to the conclusion that there is a linear relationship between target motion and hand motion. The logic of the spectral analysis was not clear to me, and the results shown in Figure 4 were not intuitive. In addition, there was no actual quantification used to make a conclusion about linearity. Thus, it was difficult to determine whether this aspect of the authors' conclusion (a critical inference for them to justify their main conclusion) was correct.

We agree with the reviewer that our approach to quantifying linearity was not explained clearly enough. We have extensively revised the section in the Results titled “Participants Used Continuous Movements to Perform Manual Tracking” to include more intuitive explanations about these analyses. We also agree that the logic relating the linearity analysis to the main conclusion was somewhat loose and will attempt to clarify them below.

Using the amplitude spectra and coherence analyses, we attempted to demonstrate that participants’ behavior was consistent with that of a linear system. By showing that behavior was linear, this would justify our use of linear systems tools for subsequent analyses. Additionally, this would provide evidence that participants’ movements responded continuously to the varying location of the target, rather than in an intermittent manner. According to linear systems theory, a linear system will always translate sinusoidal inputs into sinusoidal outputs at the same frequency, albeit potentially scaled in amplitude and shifted in phase. Thus, by showing that participants’ behavior is linear, this would suggest that they were attempting to track the target continuously.

Another point we did not explain clearly enough is the importance of spectral coherence for assessing linearity. We based this on a theory outlined by Roddey et al. (2000)^6^, which we will briefly summarize here. Roddey and colleagues showed that the entire input-output relationship of any arbitrary system responding to an arbitrary input can be broken up into three components: (1) a component that can be explained by a linear model, (2) a component that can be explained by a nonlinear model but ***not*** a linear model, and (3) a component that cannot be explained by any model (i.e., noise). Roddey and colleagues showed that the first component—the linear part of the system’s response—is proportional to the spectral coherence between the input and output (input-output coherence). Coherence is analogous to the correlation between two signals in the time domain and is therefore bounded between 0 and 1. If the coherence between an input and output is 0.5, then this implies that 50% of the system’s response can be explained by a linear model. We found that at baseline, late learning, and post-learning, the input-output coherence was high, ranging roughly between 0.5–0.8. Furthermore, by examining the coherence between different outputs, we were also able to attribute the remaining variance not accounted for by a linear model to noise, rather than to a systematic nonlinearity in participants’ behavior (see response 2.2 for more detail on this point). Therefore, a significant majority of the variance in participant’s behavior could be described by a linear model, justifying our use of linear analysis methods in subsequent analyses.

One important new analysis we have now included in the paper examines whether we could rule out the possibility that participants were employing catch-up movements to track the target (Figure 4C). This analysis involved using the frequency-dependent phase lags in the Fourier-transformed data to estimate the increase in tracking delay incurred late in learning and comparing this to previous estimates of the time cost of re-aiming. Our results suggest that participants responded to target movement too quickly to successfully deploy a re-aiming strategy. We believe this new analysis more directly addresses the plausibility of re-aiming strategies than the amplitude spectra and coherence analyses by themselves. See response 1.6 for a more detailed discussion of this new analysis.

The data shown in Figure 4B do not appear to provide clear evidence that the relationship between the cursor and the hand was approximately linear. Currently, it seems equally plausible to say that the data are approximately non-linear. Establishing a criterion for nonlinearity would be useful (e.g. shuffling a linear response for comparison).

As described in response 2.1, the coherence between target and hand movement provides an estimate for the proportion of participants’ behavior that can be explained by a linear model. The residual variance unexplained by a linear model can be attributed to either nonlinearity (as the reviewer suggests) or noise. The theory from Roddey and colleagues also provides a method to estimate the proportion of behavior that is nonlinear, which can equivalently be thought of as the proportion of the behavior that is deterministic. (This quantity can be computed as the square root of the coherence between hand movement across different trials within a block.) We found that a nonlinear model could only account for an additional 5-10% of behavioral variance compared to a linear model. This suggests that the deterministic portion of participants’ tracking behavior could mostly be explained as a linear response rather than as a systematic nonlinearity. We have included this additional analysis in lines 268-278.

However, we do agree that coherence has limited interpretability as a metric for linearity because there is no threshold for determining that a system is linear, other than exhibiting a coherence of 1 at all frequencies. We believe, however, that the fact that a substantial majority of the variability in behavior being linear offers justification for the use of linear methods in later analyses and is suggestive that participants tracked the target continuously, rather than intermittently.

Reviewer #3 raises a similar concern. The authors show the tracking strategies participants applied by investigating the relationship between hand and target movement. The linear relationship would suggest that participants tracked the target using continuous movements. In contrast, a nonlinear relationship would suggest that participants used an alternative tracking strategy. The authors only state this relationship is based on figure 4 but it seems do not provide any proof of the linearity. It would be more convincing to provide an analysis to show that the relationship is indeed linear or nonlinear.

As discussed in responses 2.1 and 2.2, we have clarified in the revised manuscript how the amplitude spectra and coherence analyses support our argument and we have also offered improved guidance as to how these measures should be interpreted.

Reviewer #2 raised similar concerns, pointing out that using linearity as a metric for mechanistic inference has limitations.– The absence of learning (errors) would present as nonlinearity.– The use of cognitive strategy could present as nonlinearity.– It doesn't seem possible to parse the two mechanisms, especially as you might expect both an increase in error at the beginning of learning and possibly an intervening cognitive strategy at the beginning of learning.

We thank the reviewer for this point. The absence of learning would not in fact present as a nonlinearity. Participants could linearly translate x-axis target motion into x-axis hand motion (consistent with baseline; no learning) or linearly translate x-axis target motion into y-axis hand motion (perfect compensation). In both cases the behavior would be linear. Errors per se would not therefore present as a nonlinearity. It may be that errors might trigger a nonlinear correction mechanism, but they also could be corrected by a linear mechanism.

We agree that the use of a cognitive strategy would likely present as a nonlinearity and we think this is a likely explanation for the low coherence (high nonlinearity + noise) early in learning. Thus, it is possible for us to parse these two possibilities.

As addressed in response 2.1, we have revised the section in the Results titled “Participants Used Continuous Movements to Perform Manual Tracking” to clarify these points.

In lines 223-225: The logic behind why coupling across axes is not nonlinear behavior seems to be missing. It's quite unclear and currently difficult to understand. It would be very helpful to spell this out too.

Coupling between the target and the hand across axes is not nonlinear; it can be described in the time domain by a simple matrix transformation. Behavioral coupling across axes is thus linear in the same way that the imposed perturbations are linear, even though they couple hand motion and cursor motion across different axes. It may also be helpful to note that the Fourier transformation into the frequency domain is also a linear operation and a linear transformation of a linear transformation is also linear. Thus, coupling across axes—whether expressed in the time domain or the frequency domain, falls within linear behavior.

We acknowledge that the intuition behind this may not be straightforward and have provided more detailed explanations of our logic in lines 226-256 and 314-323.

Given these issues, a more grounded interpretation is that linearity simply represents real-time updating. If the relationship between the cursor and the hand is nonlinear, then updating is not in real time.

We are unsure exactly what the reviewer means by ‘real-time updating’. A delay between hand and cursor would not, in fact, render the behavior nonlinear (sinusoidal input would still be translated to sinusoidal output at the same frequency but at a different phase). If the reviewer is suggesting that the cursor and hand are updated in a continuous manner, i.e. not intermittently, then we agree—this is the essence of our argument: that participants counter the mirror-reversal by using a continuous controller rather than by an intermittently applied re-aiming strategy. We are happy to further address this point if the reviewer clarifies this comment.

This was also pointed out by Reviewer #3 who pointed out that details about frequency analysis are buried deep in the methods (around line 711), especially how the hand-target coherence (shown in 4B) is calculated. It would be helpful to include some of these details in the main text. For example, it is currently very difficult to understand the relationship when from moving from Figure 4A to 4B.

We thank the reviewer for pointing out our lack of clarity in describing the methods in the Results. Indeed, the issue of coherence is not easy to intuit and is unlikely to be familiar to the majority of readers. We have attempted to better convey the intuition behind how the amplitude spectra and coherence analyses work and how to interpret the plots in Figure 4 throughout the section titled “Participants Used Continuous Movements to Perform Manual Tracking”.

3. Statistical resultsReviewer #1 points out that any of the key statistical results were buried in the main text and some were incompletely reported. Can the authors provide a table (or set of tables) of the key statistics, including at least the value of the statistical test itself and the p-value, if not also estimates of confidence on the estimates?

We have included source data files that are linked to in Figures 3, 5, and 6 that provide these statistics.

Reviewer #3 also points out that outlier rejection based on some subjects who had greatly magnified, or attenuated data seems like it might be biasing the data. Also, the outlier rejection criteria used (>1.5 IQR) seems very stringent. Furthermore, it appears there was no outlier rejection on the main experiment. It would be good to be consistent across experiments.

We thank the reviewer for pointing this out. We originally excluded a small subset of datapoints (25 out of 560) because they heavily biased group averages for the statistical analyses. However, we agree that it would be better to stay consistent across experiments, so we have reported our results without outlier rejection.

4. Experiment 2The intention for experiment 2 is to see how much training on the point-to-point task influenced adaptation mechanisms during the tracking task. Yet, this experiment still included extensive exposure to the point-to-point task. Just not as much as in experiment 1. Given this, how can an inference be cleanly made about the influence of one task on the other? Wouldn't the clean way to ask this question be to just not run the point-to-point tracking task at all?

We thank the reviewer for pointing out this concern with Experiment 2. We agree that not including any point-to-point movements at all would have enabled us to more directly assess the influence of point-to-point training on tracking performance. However, the exposure to the point-to-point task was far from “extensive” in this group. Participants only performed 15 point-to-point reaches between the early and late tracking blocks, much fewer than the 450 reaches in the main experiment (we have made this explicit in lines 431-432). Our point-to-point reaching data from experiment 1, as well as other studies (Fernandez-Ruiz et al. 2011, Taylor et al. 2014, Bond and Taylor 2015)^3,7,8^, suggest that people learn to counter large perturbations of visual feedback over many dozens of point-to-point reaches, much greater than the 15 used in Experiment 2. Furthermore, even if there were some effect of the 15 point-to-point reaches on tracking performance, this effect appears to be minimal as there was no significant difference in the off-diagonal gains between tracking trials 12 and 13 (the trials before and after the 15 point-to-point reaches) for both groups of participants (Figure 6C).

We also note that we feel it is difficult to draw any firm conclusions from this experiment about the importance of point-to-point training. Participants do appear capable of improving their performance under the mirror reversal, though there appears to be less compensation than in the corresponding group in the main experiment. This could, however, be due to less overall exposure time to the perturbation. Despite this issue, we feel that the follow-up experiment provides a valuable replication of the results of the main experiment, so we feel it is worthwhile to still include in the paper.

5. Frequency analysisThe authors state that "The failure to compensate at high frequencies.… is consistent with the observation that people who have learned to make point-to-point movements under mirror-reversed feedback are unable to generate appropriate rapid corrections to unexpected perturbations." This logic is not clear. How is this inferred based on which movement frequencies show an effect, and which do not, leading to this conclusion?

We thank the reviewer for pointing this out and agree that the link between the cited studies and our results was unclear. Although we believe these results may be related, developing this link more rigorously is actually far from straightforward (as the reviewer suggested). We have therefore decided to omit this statement from the revised manuscript, which was not germane to the main claims of our paper.

6. Clarity of logicReviewer #3 states that would be helpful if the authors could provide more background/context on their view of *de novo* learning and explanations on relationship between *de novo* learning and the adapted controller model. For example, why does the lack of aftereffects under the mirror-reversal imply that the participants did not counter this perturbation via adaptation and instead engaged the learning by forming a *de novo* controller (Line 199)? Is the reasoning purely behavioral observations, or is there a physiological basis for this assertion?

As discussed in responses 1.1 and 1.3–1.5, persistent aftereffects are a ubiquitous hallmark of adaptation. The lack or aftereffects (or at best minimal aftereffects) in the mirror-reversal group are thus inconsistent with this compensation being achieved through adaptation.

The proposed dissociation between adapting a controller and building a *de novo* controller is also supported by numerous previous studies which have found both behavioral^1,9,10^ and physiological evidence^9,11,12^ to support a dissociation between the learning mechanisms responsible for rotation versus mirror-reversal learning.

In addition, this same reviewer points out that on lines 197-199: The reason for the lack of after-effects in the mean-squared error analysis is a little vague. It took a few tries to understand the reasoning. It would be good to spell this out a little more clearly.

We thank the reviewer for pointing out this lack of clarity. The reason that aftereffects were not evident in the mean-squared error analysis is that aftereffects are typically small in magnitude and introduce relatively little error to the total mean-squared error over the course of a tracking trial. The increase in mean-squared error during early learning can largely be explained by the fact that during this block, participants’ cursors deviated far from the area of target movement. Compared to these large deviations, the errors introduced by aftereffects are relatively small. We have revised lines 152-154 to better clarify this reasoning.

7. Learning in the visuomotor rotation (VMR) condition.Reviewer #3 also shows that surprisingly, there is no measurement of aiming in the learning to VMR. Several motor learning studies (several the authors cite) show that learning in VMR is a combination of implicit and explicit. It is understood that this is not possible in the continuous tracking task, but can certainly be done in the point to point task. Is there a reason this was not done? Wouldn't this have further supported the author's claim of an existing controller?

We thank the reviewer for this suggestion. We did not assess explicit re-aiming in our point-to-point task for several reasons. One reason is this would have significantly lengthened an already quite long experiment. We preferred to use our participant’s time obtaining more extensive data in the tracking task, which was the primary focus of this study.

It is also not clear in what way collecting re-aiming data would have further supported our argument. The use of explicit strategies is well established and there seems to be little need to replicate this effect in our specific pool of participants. One possible benefit of obtaining an implicit learning measure in the point-to-point task is that it would have been possible to test whether this was correlated with the aftereffect estimated in the tracking task. We do note that the magnitude of aftereffects we observed in the tracking task was broadly consistent with experiments that have assessed implicit learning in point-to-point tasks.

A more comprehensive examination of the extent of generalization and shared mechanisms between the point-to-point and tracking conditions would certainly be interesting to examine in the future using the methods established in this paper (and building on prior studies along these lines^13^). Such questions are largely tangential to our main question in this paper, however, and we do not believe they are critical to our present conclusions.

References

1. Telgen, S., Parvin, D. and Diedrichsen, J. Mirror Reversal and Visual Rotation Are Learned and Consolidated via Separate Mechanisms: Recalibrating or Learning *de novo*? *J. Neurosci.* 34, 13768–13779 (2014).

2. Wilterson, S. A. and Taylor, J. A. Implicit visuomotor adaptation remains limited after several days of training. *bioRxiv* 711598 (2019) doi:10.1101/711598.

3. Fernández-Ruiz, J., Wong, W., Armstrong, I. T. and Flanagan, J. R. Relation between reaction time and reach errors during visuomotor adaptation. *Behav. Brain Res.* 219, 8–14 (2011).

4. McDougle, S. D. and Taylor, J. A. Dissociable cognitive strategies for sensorimotor learning. *Nat. Commun.* 10, 40 (2019).

5. Huberdeau, D. M., Krakauer, J. W. and Haith, A. M. Practice induces a qualitative change in the memory representation for visuomotor learning. *J. Neurophysiol.* 122, 1050–1059 (2019).

6. Roddey, J. C., Girish, B. and Miller, J. P. Assessing the performance of neural encoding models in the presence of noise. *J. Comput. Neurosci.* 8, 95–112 (2000).

7. Taylor, J. A., Krakauer, J. W. and Ivry, R. B. Explicit and implicit contributions to learning in a sensorimotor adaptation task. *J. Neurosci. Off. J. Soc. Neurosci.* 34, 3023–3032 (2014).

8. Bond, K. M. and Taylor, J. A. Flexible explicit but rigid implicit learning in a visuomotor adaptation task. *J. Neurophysiol.* 113, 3836–3849 (2015).

9. Gutierrez-Garralda, J. M. *et al.* The effect of Parkinson’s disease and Huntington’s disease on human visuomotor learning. *Eur. J. Neurosci.* 38, 2933–2940 (2013).

10. Lillicrap, T. P. *et al.* Adapting to inversion of the visual field: a new twist on an old problem. *Exp. Brain Res.* 228, 327–339 (2013).

11. Schugens, M. M., Breitenstein, C., Ackermann, H. and Daum, I. Role of the striatum and the cerebellum in motor skill acquisition. *Behav. Neurol.* 11, 149–157 (1998).

12. Maschke, M., Gomez, C. M., Ebner, T. J. and Konczak, J. Hereditary cerebellar ataxia progressively impairs force adaptation during goal-directed arm movements. *J. Neurophysiol.* 91, 230–238 (2004).

13. Abeele, S. and Bock, O. Transfer of sensorimotor adaptation between different movement categories. *Exp. Brain Res.* 148, 128–132 (2003).

14. Miall, R. C. and Jackson, J. K. Adaptation to visual feedback delays in manual tracking: evidence against the Smith Predictor model of human visually guided action. *Exp. Brain Res.* 172, 77–84 (2006).

15. Yamagami, M., Peterson, L. N., Howell, D., Roth, E. and Burden, S. A. Effect of Handedness on Learned Controllers and Sensorimotor Noise During Trajectory-Tracking. *bioRxiv* 2020.08.01.232454 (2020) doi:10.1101/2020.08.01.232454.